



**Manuscript Title**
Monitoring the variations of evapotranspiration due to the land use/cover changes in a
semiarid shrubland
Tingting Gong, Huimin Lei, Dawen Yang, Yang Jiao, Hanbo Yang
State Key Laboratory of Hydroscience and Engineering, Department of Hydraulic
Engineering, Tsinghua University, Beijing, 100084, China
**Correspondence to**: Huimin Lei (leihm@tsinghua.edu.cn)
**Abstract**
Evapotranspiration ($E_T$) is an important process in the hydrological cycle, and
vegetation change is a primary factor that affects $E_T$. In this study, an attempt is made
to analyze the annual and inter-annual characteristics of $E_T$ using continuous
observation data from eddy-covariance (EC) measurements over four periods (1$^{st}$ July
2011 to 30$^{th}$ June 2015) at a study site located in the Mu Us Sandland of China.
Normalized vegetation index (NDVI) was demonstrated as the predominant factor that
influences the seasonal variation in $E_T$. Normalization method was adopted to exclude
the effects of potential evapotranspiration ($E_{TP}$) and soil water stress ($f_s$) on $E_T$.
Vegetation phenological process was validated to have a remarkable positive effect on
normalized $E_T$ in a rate of 1.86 (the slope of normalized $E_T$ per NDVI) along with
vegetation greening. Both on the land degradation process and vegetation rehabilitation
process, $E_T$ and normalized $E_T$ increased. We discussed several possibilities that might
lead to the increase. Our work may promote our knowledge about the characteristics of





$E_T$ of the mix land use/cover condition (sparse shrubland and grassland) in the fragile
ecosystem of Mu Us Sandland.
**Key words**: evapotranspiration; vegetation phenology; land use/cover change; eddy
covariance; Mu Us Sandland



1 Introduction

Arid and semiarid biomes cover about 40% of the Earth's terrestrial surface (Fernández, 2002). Previous studies have shown that more than 50% of precipitation ($P$) is consumed by $E_T$ (Yang et al., 2007; Liu et al., 2002), and that the ratio of $E_T/P$ could increase to even 90% or more in semiarid and arid areas (Mo et al., 2004; Glenn et al., 2007). Therefore, a slight change in $E_T$ would have significant influences on water cycle in arid and semiarid regions. $E_T$ is not only affected by climatic factors (e.g., radiation, temperature, and relative humidity), but also affected by vegetation conditions (Tian et al., 2015; Wang et al., 2011; Piao et al., 2006; Mackay et al., 2007). As such, there has been an important push to understand how $E_T$ responds to vegetation conditions in these regions.

Vegetation change mainly integrates the phenological change (temporal) and land use/cover change (spatial). The phenological change reflects the response of plants to climate change (vegetation greening and browning process) (Ge et al., 2015), which actively controls $E_T$ process through internal physiology such as stomatal conductance (Pearcy et al., 1989) and stomatal numbers and sizes (Turrell, 1947). In general, transpiration is in direct proportion to stomatal conductance at the leaf-level scale (Leuning et al., 1995). Meanwhile, at canopy scale, $E_T$ is positively proportional to surface conductance that is an integration of stomatal conductance and leaf area (Ding et al., 2014). Thus, as a good indicator of vegetation phenological change, many studies have found that $E_T$ was positively related to vegetation index such as Normalized difference vegetation index (NDVI) (Gu et al., 2007). Land use/cover change influences



$E_T$ by means of modifying vegetation species with different transpiration rates,
radiation transfers within canopy (Martens et al., 2000; Panferov et al., 2001),
topography (Lv et al., 2006), albedos (Zeng et al., 2009), soil texture (Maayar and Chen,
2006), litter coverage (Wang, 1992), and biological soil crusts (Yang et al., 2015, Fu et
al., 2010; Liu, 2012; Eldridge and Greene, 1994). These complex processes result in no
consensus about the effects of land use/cover changes on $E_T$. For example, during the
land degradation process, some researchers found that warming air temperature
increase was the dominant cause to make $E_T$ increase (Zeng and Yang, 2008; Li et al.,
2000; Deffema and Freire, 2001). In contrary, $E_T$ was found to decrease along with
deforestation because of less transpiration (Snyman, 2001; Souza and Oyama, 2011) or
higher albedos (Zeng et al., 2002). Moreover, no differences of $E_T$ during land
degradation was also reported (Hoshino et al., 2009). Therefore, the impacts of land
use/cover changes on $E_T$ still deserve further investigations.
The Mu Us Sandland is a semiarid shrubland ecosystem at the northern margin of
the Loess Plateau in China, covering an area of only 40,000 km$^2$ (Dong and Zhang,
2001). The region is ecologically fragile (Yang et al., 2007). Shortage of water is the
critical limiting factor on vegetation, and drought-enduring vegetation are prevailed as
a result of common droughts (Wang et al., 2002; Wu, 2006). There are at least 117 shrub
and semi-shrub species have been found within the Mu Us Sandland (Dong and Zhang,
2001). In such arid and semiarid ecosystem, sand dunes and biological soil crusts (BSCs)
are commonly observed (Gao et al., 2014; Yang et al., 2015; Li and Li, 2000; Liu, 2012).
Due to the exists of BSCs (Yang et al., 2015; Fu et al., 2010; Liu, 2012) and dry sand





layers (Wang et al., 2006; Feng, 1994; Liu et al., 2006; Yuan et al., 2007), soil
evaporation have been effectively retained, therefore, the Mu Us Sandland holds
abundant groundwater (Li and Li, 2000). During the past decades, rapid land use/cover
change has taken place in this region due to agricultural reclamation (Wu et al., 1997;
Wu and Ci, 2002; Wang et al., 2009; Ostwald and Chen, 2006; Zhang et al., 2007),
leading to a dramatic change in vegetation conditions. With respect to the specific
question of whether land use/cover changes will lead to increases in $E_T$ or not, a
continuous measurement of $E_T$ under different land use/cover conditions is needed in
this region.

Three methods were usually employed to assess the impacts of land use/cover

change on $E_T$: numerical models, paired comparative approaches and the in situ filed
observations. In these methods, numerical models are widely used (Twine et al., 2003;
Feddema et al., 2005; Kim et al., 2005; Li et al., 2009; Cornelissen et al., 2013; Mo et
al., 2004). However, model parameterization of vegetation condition is a big challenge
as the complex underlying mechanisms mentioned above cannot be completely
considered in the models. Therefore, the simulated impacts of land use/cover change
on $E_T$ is highly dependent on the model parameterizations, and the resulting conclusions
may be doubtful (Cornelissen et al., 2013; Li et al., 2009). Paired comparative approach
is often considered as the best method, but it is difficult to find two similar medium and
large-sized sites with different land use/cover conditions (Li et al., 2009; Lorup et al.,
1998). In situ observation is also a widely used method for long-term land-atmosphere
exchange measurements. However, the land use/cover conditions at the sites are usually





stable, and only the responses of $E_T$ to vegetation phenology change can be studied. For
example, the characteristics of $E_T$ under grassland (Zhang et al., 2005), mixed
plantation (cork oak, black locust and arborvitae) (Tong et al., 2014), vineyard (Li et al.,
2015) and grazed steppe (Chen et al., 2009; Vetter et al., 2012). To our knowledge, there
is little learned of $E_T$ under native sparse shrubland and continuous filed observations
under land degradation and vegetation rehabilitation conditions are not documented.
Our study site is at the edge of the Mu US Sandland. Coincidentally, land
degradation and vegetation rehabilitation has occurred at this site, which provides us a
unique opportunity to study the effects of land use/cover change on $E_T$. Based on the 4-
year measurements of $E_T$ by eddy covariance technique, this study analyzed the
seasonal and inter-annual variations of $E_T$, and discussed the possible reasons for the
responses of $E_T$ to land use/cover changes.

2 Materials and methods
2.1 Site description
The study was carried out at Yulin flux site (N 38 ′26 ′, E 109 ′47 ′, 1233 m), which
was established in June 2011 and is in a landform transition zone change from Mu Us
Sandy land to north Shaanxi Loess Plateau (Fig. 1). The study site is in a temperate
semiarid continental temperate monsoon climate. According to the long-term climate
data (1951-2012) from a meteorological station in Yulin (Fig. 1), the annual
precipitation varies from 235 mm to 685 mm, with a mean of 402 mm, and more than
50% of annual precipitation is falling in the monsoon season (July-September). The





mean annual air temperature is 8.4 °C during the past 61 years. The dominant soil type
is sand (98% sand) (saturated soil water content: 0.43 $m^3m^{-3}$, field capacity: 0.16 $m^3m^{-}$
$^3$, residual moisture content: 0.045 $m^3m^{-3}$). There are widely distributed fixed dunes and
semi-fixed dunes around the site, and the depth of dry sand layer is 10 cm (Wang et al.,
2006). The mean groundwater depth of our study site from 1$^{st}$ July 2011 to 30$^{th}$ June
2015 was 3.5 m.

[Figure 1 is to be inserted here]

The experimental site is mainly covered with mixed vegetation, one kind of

vegetation is the native drought-enduring shrubs with low water demands such as
*Artemisia ordosica* and *Salix psammophila* (Fig.2a); the other kind is the sparse
grassland that mainly distributed at the bottom of sand dunes because of better soil
moisture condition (Lv et al., 2006). They constitute the dominant vegetation in Mu Us
Sandland (An et al., 2011) and are adapted well to semiarid and arid sites. According
to our observations around the flux tower on 14$^{th}$ June 2011, the maximum root depth
of the shrubs was approximately 160 cm. Xiao et al. (2005) reported that the growing
season of *Artemisia ordosica* and *Salix psammophila* spanned from late April to late
September. Therefore, we defined the period from 1$^{st}$ May to 30$^{th}$ September as
vegetation growing season for data analysis in this study. On 15$^{th}$ August 2011 and 7$^{th}$
September 2011, we did surveys about the vegetation coverage with randomly selected
7 samples around the flux tower (5 × 500 cm × 500 cm and 2 × 1000 cm × 1000 cm),
and found that the vegetation coverage was 28.2% and 27.9%, respectively.

[Figure 2 is to be inserted here]





At the end of June 2012, the land use/cover condition around the east area of flux
tower began to be changed by farmers (the natural vegetation including the leaves and
branches was cut-off, and the sand dunes were bulldozed) (Fig. 2c), converting part of
the natural vegetated land to bare soil, with the planning of planting potatoes in the
future. As time goes on, natural grass grew out gradually in the bare land before planting
potatoes. Thus, our study period can be divided into four periods according to the land
use/cover conditions: Period I (1$^{st}$ July 2011 to 30$^{th}$ June 2012) was the natural land
use/cover condition (i.e., mixed sparsely distributed shrubs and grass) (Fig.2a and
Fig.2b); Period II (1$^{st}$ July2012 to 30$^{th}$ June 2013) was the transitional period with land
use/cover condition starting to change with partial natural vegetation being cut-off and
sand dunes being bulldozed; Period III (1$^{st}$ July 2013 to 30$^{th}$ June 2014) was the period
when the land use/cover condition constituted two parts, one was the natural vegetation
zone and the other was the bare soil zone (Fig.2c); Period IV (1$^{st}$ July 2014 to 30$^{th}$ June
2015) was the period when the bare soil zone was gradually covered by re-growing
grass (Fig.2d).

2.2 Measurements
2.2.1 Eddy covariance system
Net exchange of water vapor between atmosphere and canopy at this site is
measured by the eddy-covariance (EC) flux measurements, which assess the fluxes of
land-atmosphere (such as water and energy) (Baldocchi et al., 2001). The data are
essential for the estimation of the water and energy balance (Franssen et al., 2010). At



our site, EC system is installed at a height of 7.53 m above the ground surface, using
CSAT3 three-dimensional sonic anemometers (Campbell Scientific Inc., Logan, UT,
USA) for wind and temperature fluctuations measurements and a LI-7500A open-path
infrared gas analyzer (LI-COR, Inc., Lincoln, NE, USA) for water vapor content
measurement.
2.2.2 Other measurements
Net radiation ($R_n$) is measured by a net radiometer (CNR-4; KIPP&ZONEN, Delft,
the Netherlands), including four radiometers measuring the incoming and reflected
short-wave radiation ($R_S$), and incoming and outgoing long-wave radiation ($R_L$). Wind
speed and direction (05103, Young Co. Traverse City, MI, USA) are measured at 10 m
above the ground surface. Precipitation ($P$, mm) is recorded with a tipping bucket rain
gauge (TE525MM; Campbell Scientific Inc., Logan, UT, USA) installed at a height of
0.7 m above the ground surface. Air temperature ($T_a$) and relative humidity ($R_H$) are
measured by a temperature and relative humidity probe (HMP45C; Campbell Scientific
Inc., Logan, UT, USA) at a height of 2.6 m above the ground surface. Soil water content
($\theta$) is measured by Time Domain Reflectometry (TDR) sensors (CS616; Campbell
Scientific Inc., Logan, UT, USA), soil temperature ($T_s$) is measured by thermocouples
(109; Campbell Scientific Inc., Logan, UT, USA), and soil heat flux ($G$) is measured
by heat flux plates (HFP01SC; Campbell Scientific Inc., Logan, UT, USA) at a depth
of 0.03 m below the ground surface. These ground variables ($G$, $\theta$, $T_s$) are measured
beneath the surface at two profiles (1) a plant canopy patch and (2) a bare soil patch. $\theta$
and $T_s$ are measured at depths of 5, 10, 20, 40, 60, 80, 120 and 160 cm below the ground





surface. Groundwater table is measured by an automatic sensor (CS450-L; Campbell
Scientific Inc., Logan, UT, USA), which is installed in a groundwater well close to the
tower.

2.3 Data and methodology
2.3.1 Flux data processing

The half-hourly latent heat flux ($\lambda E_T$) data were calculated by EddyPro software

(www.licor.com/eddypro) based on the raw data collected from the EC technique, and
it is widely used because it is comprehensive, freely available and use-friendly (Fratini
et al., 2014). The calculated half-hourly flux datasets were filtered for spikes,
instrument malfunctions, and poor quality, according to the following criteria (Papale
et al., 2006): (1) incomplete half-hourly measurement, mainly caused by power failure
or instrument malfunction; (2) rainy events; and (3) outliers caused by occasional spikes
for unknown reasons. The ratios of data removed through this procedure are 17.3% in
Period I, 20.2% in Period II, 16.5% in Period III and 18.6% in Period IV.

Daily averaged flux data were calculated by firstly gap-filled half-hourly data.

Linear interpolation was used to fill gaps less than 1-h by calculating an average of the
values immediately before and after the data gap. Larger gaps (gaps more than 1-h but
less than 7-days) in flux data were replaced by average values using mean diurnal
variation (MDV) methods (Falge et al. 2001). This method is adopted by FLUXNET
for standardized gap-filling. We found that the daily $\lambda E_T$ had the best correlation with
daily available energy ($R_n - G$) rather than other environmental variables such as vapor





pressure deficit (VPD) and NDVI. Therefore, for some large gaps more than 7-days and
less than 15 days in daily $\lambda E_T$, we fitted the relationship between daily $\lambda E_T$ and daily
available energy flux $(R_n - G)$ in each period. Then we used the fitted function $f$ to
estimate the daily $\lambda E_T$ of gaps. We chose the function $f$ with the highest coefficient
of correlation $(R)$ in each period (Yan et al., 2013). The function $f$ of each period was
$\lambda E_T = 0.0014 (Rn - G)^2 + 0.0746 (Rn - G) + 10.69$ (Period I, $R = 0.77$), $\lambda E_T =$
$0.0012(Rn - G)^2 + 0.0559(Rn - G) + 17.69$  (Period II, $R = 0.67$), $\lambda E_T =$
$0.0014(Rn - G)^2 + 0.16(Rn - G) + 13.244$  (Period III, $R = 0.75$), and $\lambda E_T =$
$0.0015(Rn - G)^2 - 0.0834(Rn - G) + 25.868$ (Period IV, $R = 0.69$), respectively.
Large gaps of more than 7-days did occur in the winter.

2.3.2 Footprint model
In order to determine the contributing source area of flux at our site, scalar flux
footprint model proposed by Heish et al. (2000) was used. The analytic model
accurately described the relationship between footprint, observation height, surface
roughness, and atmospheric stability. The footprint fetch $F_f$ is calculated by,
$F_f / Z_m = D/(0.105 \times k^2) \, Z_m^{-1} |L|^{1-Q} Z_u^Q$     (1)
where $k$ is the von Karman constant $(=0.40)$, $D$ and $Q$ are the similarity constants
(stable conditions: $D = 0.28$, $Q = 0.59$; near neutral and neutral conditions: $D = 0.97$, $Q$
$= 1$; unstable conditions: $D = 2.44$, $Q = 1.33$), $L$ is the Obukhov Length, $Z_m$ is the
height of wind instrument $(=7.53 \text{ m})$, $Z_u$ is defined as (Heish et al, 2000),
$Z_u = Z_m(\ln(Z_m/Z_0) - 1 + Z_m/Z_0)$     (2)



where $Z_0$ is the height of momentum roughness (0.05 m).

2.3.3 Methods of analyzing controlling factors on $E_T$
It is generally recognized that potential evapotranspiration ($E_{TP}$), vegetation
condition and soil water content are the three main factors controlling $E_T$ (Lettenmaier
and Famiglietti, 2006; Chen et al., 2014). In order to decouple the effect of vegetation
change from the integrated effects of these three factors on $E_T$, we used a simple
equation which is similar with the FAO single crop coefficient method (Irrigation and
Drainage Paper No. 56 (FAO-56)) and is expressed as,
$E_T = E_{TP} \times f_v(\text{vegetation}) \times f_s(\text{soil water})$       (3)
where $f_v(\text{vegetation})$ represents the effect of vegetation change on $E_T$, and
$f_s(\text{soil water})$ represents the effect of soil water content on $E_T$. By transforming the
Eq.3, $f_v(\text{vegetation})$ can be expressed as,
$f_v(\text{vegetation}) = E_T / [E_{TP} \times f_s(\text{soil water})]$       (4)
where $f_v(\text{vegetation})$ can also be regarded as the normalized $E_T$ which eliminates the
effects of atmospheric and soil water content. $E_{TP}$ (mm day$^{-1}$) was estimated by the
following equation (Maidment, 1992) which is a modification of Penman equation,
$E_{TP} = \frac{\Delta}{\Delta+\gamma}(R_n - G) + \frac{\rho_a c_p / r_a}{\Delta+\gamma}\frac{\text{VPD}}{\lambda}$       (5)
where the units of $R_n$ and G are mm d$^{-1}$; $\rho_a$ is the air density ($= 3.486 \frac{P_a}{275+T}$, kg m$^{-3}$,
where $P$ is the atmospheric pressure in kPa and $T$ is air temperature in degrees Celsius);
$c_p$ is the specific heat of moist air (=1.013 kJ kg$^{-1}$ ℃$^{-1}$); $\Delta$ is the slope of saturation
vapor-pressure-temperature curve (kPa ℃$^{-1}$); $\gamma$ is the psychrometric constant (kPa ℃$^{-}$





$^1$); VPD is the difference of the mean saturation vapor pressure ($e_s$, kPa) and actual
vapor pressure ($e_a$, kPa); $U_2$ is the daily wind speed at a height of 2.0 m (m s$^{-1}$), which
was simulated by the wind speed at the height of 10.0 m (m s$^{-1}$),
$$U_2 = U_{10} \frac{4.87}{\ln(67.8*10-5.42)} \tag{6}$$
$r_a$ is the aerodynamic resistance, which was calculated as (Penman, 1948; 1963),
$$r_a = \frac{4.72[ln\left(\frac{Z_h}{Z_0}\right)][ln\left(\frac{Z_h}{Z_0}\right)]}{1+0.536U_2} \tag{7}$$
where $Z_h$ is the height at which meteorological variables are measured (2 m), and $Z_0$
is the aerodynamic roughness of surface (0.00137 m) (Penman, 1948; 1963).
The effects of soil water content on $E_T$ can be described in three stages (Idso et al.,
1974), stage 1: the soil water is enough to satisfy the potential evaporation rate ($f_s$=1);
stage 2: the soil is drying and water availability limits $E_T$ (0<$f_s$<1); and stage 3: the soil
is dry and evaporation can be considered negligible ($f_s$=0). We used daily soil water
content of the root depth ($\theta_r$) to estimate $f_s$ by the following expression (Hu et al.,

2006),

$$f_s = \begin{cases} = 1 & \theta_r > \theta_k \\ = 0 & \theta_r < \theta_w \\ = \frac{\theta_r - \theta_w}{\theta_k - \theta_w} & \theta_w \leq \theta_r \leq \theta_k \end{cases} \tag{8}$$
where $\theta_w$ is the wilting value, $\theta_k$ is the stable field capacity which is considered to
be equivalent to 60% of the field capacity (Lei et al., 1988; Wang et al., 2008). $\theta_r$ (m$^3$
m$^{-3}$) is the mean soil water content from surface to the depth of 160 cm (root zone) and
was calculated by measured soil water contents at different depths,
$$\theta_r = \frac{0.5[10\theta_5+15\theta_{10}+30\theta_{20}+40(\theta_{40}+\theta_{60})+60\theta_{80}+80\theta_{120}+40\theta_{160}]}{160} \tag{9}$$
Site-averaged soil water content of each depth ($\theta_i$; $i$ =5, 10, 20, 40, 60, 80, 120,





and 160 cm) was calculated by taking a weighted average of the measured values in the
canopy and bare surface patches,
$\theta_i = M \times \theta_{i,c} + (1 - M) \times \theta_{i,b}$         (10)
where $\theta_{i,c}$ and $\theta_{i,b}$ refer to the measured soil water content of canopy patch and bare
soil patch at the depth of $i$ cm, respectively; M is the monthly vegetation coverage of
undisturbed zone, and it was calculated by monthly Normalized Difference Vegetation
Index (NDVI) values (Gutman and Ignatov, 1998),
$M = (NDVI - NDVI_{min})/(NDVI_{max} - NDVI_{min})$         (11)
where $NDVI_{max}$ is the maximum value (0.8 in this study); $NDVI_{min}$ is the minimum
value (0.05 in this study) (Gutman and Ignatov, 1998). The calculated monthly M (27.6%
and 24.2%) was consistent with the measured vegetation coverage in August 2011
(28.2%) and September 2011 (27.9%) at our study site.
In this study, vegetation phenology is represented by Moderate Resolution Imaging
Spectroradiometer (MODIS)-NDVI data when the land use/cover condition is fixed.
NDVI is sufficiently stable to reflect the seasonal changes of any vegetation (Huete
et.al, 2002). Higher NDVI usually represent greater photosynthetic capacity (greenness)
of vegetation canopy (Gu et al., 2007; Tucker, 1979). The daily MODIS/Terra and
MODIS/Aqua Surface Reflectance (at 250m) data within the footprint source area were
chosen to calculate NDVI. The Surface Reflectance data of MODIS/Terra (MOD09GQ)
and MODIS/Aqua (MYD09GQ) were downloaded from reverb
(http://reverb.echo.nasa.gov). MODIS Reprojection Tool (MRT) (Kalvelage and
Willems, 2005) was used to reject the daily Surface Reflectance data to the Universal





Transverse Mercator (UTM). The calculation of NDVI is based on its definition,
$$\mathrm{NDVI_{Terra\ or\ Aqua}} = \frac{\mathrm{NIR-VIS}}{\mathrm{NIR+VIS}}$$    (12)
where $\mathrm{NDVI_{Terra}}$ and $\mathrm{NDVI_{Aqua}}$ are the NDVI values calculated from MODIS/Terra
and MODIS/Aqua reflectance data, respectively; NIR is the spectral response in the
near-infrared band (857 nm); VIS is the visible red radiation band (645 nm). In order to
eliminate the poor quality data values, the calculated NDVI data series stack needs to
be firstly filtered to remove anomalous hikes and drops (Lunetta et al., 2006). Hikes
and drops were eliminated by removing the values that suddenly decreased or increased,
and then smoothing spline was used to produce a smoother profile. In this study, daily
NDVI value was averaged from $\mathrm{NDVI_{Terra}}$ and $\mathrm{NDVI_{Aqua}}$.
Theoretically, land use change can be evaluated by comparing the land use maps in
two different periods. However, the transient land use maps are unavailable at our site.
Therefore, we separated the study area within the footprint area into two zones: we
assigned the undisturbed zone without any land use/cover change as zone A, and
assigned the disturbed zone with land use/cover change as zone B. In zone A, vegetation
condition change included only vegetation phenological change; however, in zone B,
there were not only vegetation phenological change but also land use/cover change. By
assuming that the phenological changes caused by climate in the two zones are same,
we defined an indicator ($D_{\mathrm{lu}}$) to be the measure of land use/cover change:
$$D_{lu} = M_{\mathrm{A}} - M_{\mathrm{B}}$$    (13)
where, $M_{\mathrm{A}}$ and $M_{\mathrm{B}}$ are the vegetation coverage of zone A and zone B, respectively.



3 Results
3.1 Footprint and energy balance closure

Based on the footprint model, we got the half-hourly scatter data of footprint fetch

(Eq. (2)), and according to the wind rose (Fig. 3a), the prevailing wind direction in this
site were northwest and southeast, so we chose an ellipse to enclose the scatters and
simulated the footprint (Fig. 3b). The long axis of the simulated ellipse is 1682 m, and
the short axis is 1263 m. There were 93% half-hourly flux data within the ellipse under
unstable conditions. We measured the boundary of zone B in October 2013 when the
land use/cover condition in zone B had stopped to change (Fig.3b). There were 11 pixels
(250 m $\times$ 250 m) in zone A, while there are 19 pixels (250 m $\times$ 250 m) in zone B, and
thus in the following part of calculating the weight-averaged NDVI ($NDVI_w$) within the
footprint fetch, we chose the weighted coefficient as $\beta = 11/(11 + 19)$.

[Figure 3 is to be inserted here]

In order to validate EC measurements and examine the quality of flux data, we used

daily data in period I to conduct the linear regression between available energy ($R_n$ —
$G$) and the sum of surface fluxes ($\lambda E_T + H$). The linear regression yielded a slope of
0.87, an intercept of -1.42 W m$^{-2}$, and $R^2$ of 0.82. These indicators indicated that the
measurements at our experimental site provided reliable flux data, and that the EC
measurements underestimated the sum of surface fluxes to the extent of 13%. A lot of
researchers have investigated the energy imbalance (Barr et al., 2006; Wilson et al.,
2002; Franssen et al., 2010), and there is a consensus that it is difficult to examine the
exact reasons leading to the imbalance.




### 3.2 Characteristics of environmental variables

A brief summary of the key environmental variables will be presented in this section. Monthly $D$s was much higher than the normal value of 1954-2014 except in July and September. The highest value of monthly $D$s was in May (299.5 h) and the lowest was in February (206.6 h). Seasonal characteristics of $T$a showed a highly similar trend with the normal, and the differences were less than 1 ℃ except in July, January and March. The highest value of monthly $T$a was in July (22.1 ℃) and the lowest was in December (-8.1 ℃). The values of $R_H$ showed almost lower than the normal, especially in March and April. The highest $R_H$ was in September (65.4%) and the lowest was in March (35.1%). The seasonal distributions of $P$ were consistent with the normal, and 89.7% of $P$ happened in the growing season. The value of $P$ in July was the highest (120.5 mm) and in January was the lowest (0.3 mm)

[Figure 4 is to be inserted here]

The inter-annual characteristics of daily $T$a, $R_H$, $D$s, $\theta_r$, groundwater level (GWL) and total $P$ in the growing season of each period were listed in Tab.1.

[Table 1 is to be inserted here]

The values of $T$a, $R_H$, $P$ and $\theta_r$ in the growing season of Period IV were the lowest compared with other three periods. Period I~III are all wet year, while Period IV was the dry year. The values of $\theta_r$ in Period I~III were basically the same, however, $\theta_r$ decreased by 0.0113 $m^3$ $m^{-3}$ in Period IV. The mean GWL in Period III was the shallowest.





### 3.3 Seasonal variations in $E_T$ due to climate variability


Seasonal curve of $E_T$ in each year had a single peak value (Fig.5a), with the higher


$E_T$ appearing mostly in the growing season while the lower appeared in the non-


growing season. The daily $E_T$ was in a range from 0.0 mm day$^{-1}$ to 6.8 mm day$^{-1}$ during


the four periods, the highest $E_T$ appeared on 22$^{th}$ June 2013. The highest $E_T$ appeared


at the day after a continual rainfall event start from 19$^{th}$ June 2013 to 21$^{th}$ June 2013


(90.3 mm), $E_T$ rates normally increase rapidly after rainfall events. The lowest $E_T$ was


on 28$^{th}$ November 2012, which was in the frozen period (late November to early March


in our study site). In rainy days, $E_{TP}$ (Fig.5b) was much lower due to lower net radiation


and air temperature. $E_{TP}$ was in the range of 0.2 mm day$^{-1}$ that appeared in December


2011 to 17.9 mm day$^{-1}$ that appeared in September 2013.


[Figure 5 is to be inserted here]


Seasonal NDVI curve with natural land use/cover condition (in zone A during


Period I~IV and in zone B during Period I) represented the process of natural vegetation


phenology and it had one single peak value in each year (Fig. 5c). In early May,


seasonal NDVI curve began to increase and native vegetation began to enter the


growing season and reached to the maximum value (0.27±0.01) in July or August. In


winter, daily NDVI basically stayed at a constant value (0.13±0.01). $f_s$ (Fig. 5d)


increased rapidly in response to rainfall events of more than 5 mm a day, and also


decreased rapidly one or two days later after rainfall events. During late November to


early March, there was a frozen period in this site, and soil water content was below the


wilting point. The groundwater level fluctuated obviously in monsoon season (July to




September) and mildly in winter (December to February).
The relationships between $E_T$ and the three factors ($E_{TP}$, $NDVI_w$ ($NDVI_w =$
$NDVI_A \times \beta + NDVI_B \times (1 - \beta)$), $f_s$) were analyzed and were shown in Fig. 6 (a, b, c)
by daily data in Period I. Because in Period I, the land use/cover condition within
footprint was undisturbed. Data in rainy days was removed, because in rainy days, $E_T$
was gap-filled instead of actual measured.

[Figure 6 is to be inserted here]

In order to figure out the major seasonal factor that control $E_T$ at our study site,
significant T-test was calculated to evaluate the degree of correlation. The linear
correlations between $E_T$ and the three factors both passed the 95% $t$-test confidence
level. The determination coefficient ($R^2$=0.52) between $E_T$ and $NDVI_w$ was the largest,
indicating that NDVI was a dominant factor that controlling the daily variations of $E_T$.
To better quantify the effects of phenological process on $E_T$, daily normalized $E_T$ ($f_v$)
and $NDVI_w$ in Period I were analyzed (Fig.7a).

[Figure 7 is to be inserted here]

Positive linear regression was found between $f_v$ ($f_v = E_T/(E_{TP} \times f_s)$) and
$NDVI_w$ (Fig.7a). The slope of linear regression was used to evaluate the controlling
degree between normalized $E_T$ and vegetation phenological process. The positive
regression stated the direct positive relationship between $NDVI_w$ and normalized $E_T$,
indicating that when $NDVI_w$ increases one unit, it will contribute normalized $E_T$ to
increase about 1.86 units.





3.4 Inter-annual variations in $E_T$ due to land use/cover changes
During the four periods, in zone A, the NDVI values of each period were basically
the same because the land use/cover condition was not changed. While in zone B, the
peak values of NDVI firstly declined from 0.28 to 0.15 (Period I to Period III) due to
the land use/cover condition changed to bare soil and then increased to 0.22 due to the
grass recovery (Figure 5(c)). An interesting phenomenon was found accompanied by
the changing process of land use/cover condition: $E_T$ in the growing season of each
period was gradually observed to be increasing (Tab.2). $E_T$ in Period IV increased
strongly even with less precipitation, because a mass of soil water and ground water
was consumed to satisfy the $E_T$ demand (Fig.5e).

[Table 2 is inserted to be here]

Compared to Period I with natural land use/cover condition, $D_{lu}$ of Period II and
Period III gradually increased and $D_{lu}$ of Period IV decreased. Taking August in each
period as an example, in August of Period I, $D_{lu}$ was 0.2%, while in August of Period
II, Period III and Period IV, $D_{lu}$ were 2.9%, 12.6% and 8.6%, respectively. In order to
eliminate the influence of vegetation phenological change on $E_T$, we chose the growing
season of each period to analyze the correlations between normalized $E_T$ and $D_{lu}$.
Quantitative results of the relationship between $D_{lu}$ and normalized $E_T$ ($f_v$) are
shown in Fig.7b. According to the dynamic path showed in Fig.7b, when the natural
vegetation in Zone B was cut-off, the fixed and semi-fixed sand dunes were bulldozed,
the BSCs and dry sand layers were disappeared (Period I~III), normalized $E_T$ (i.e., $f_v$)
increased and was more evident in Period III (from 78.5 to 88.1). When the land



use/cover condition of zone B gradually changed to sparse grassland due to the self-
restoring capacity of nature, normalization $E_T$ ($f_v$) increased more significantly (from
88.1 to 111.3).

4 Discussion
4.1 Implications of the impacts of phenological change on $E_T$
The correlations between $E_T$ and its controlling factors infer that at our
experimental site, NDVI was the predominant factor that influence the seasonal
variation on $E_T$. The positive linear relationship between normalized $E_T$ and NDVI
suggests that transpiration is mainly controlled by the stomatal conductance and the
number of stomata, which is proportional to leaf area (Pearcy et al., 1989; Turrell, 1947),
rather than the atmospheric water demand represented by $E_{TP}$.
Various studies have tested the relationships between phenological change and $E_T$,
and these results generally showed consistent and positive linear relationships (Nouri
et al., 2014; Rossato et al., 2005; Duchemin et al., 2006; Glenn et al., 2008). However,
with different vegetation species, phenological change have effects on $E_T$ in different
degrees. Loukas et al. (2005) have analyzed the relationships between NDVI and $E_T$ in
Greece, and relative strong regressions were found in forested sites ($R^2$=0.78). Kondoh
and Higuchi (2001) investigated the correlation between NDVI and $E_T$ in a grass-
covered site at the university of Tsukuba, and a very high determination coefficient
($R^2$=0.92) was showed to reveal the strong control of phenological change on $E_T$. Nouri
et al. (2014) have analyzed the relationships between NDVI and $E_T$ in forests and
grasses, and they found that determination coefficient of forests ($R^2$=0.94) was higher





than the grassland ($R^2$=0.88). Chong et al. (2007) have found a strong relationship
between NDVI and $E_T$ in forests and moist savanna of Africa. Thus, we speculate that,
for high dense vegetated ecosystems, phenological change might have a strong and
significant control on $E_T$. However, in low vegetation cover condition such as sparse
shrubland in this study, the relationship between $E_T$ and seasonal vegetation change is
thus positive but relative weak.

4.2 Possible reasons for the effects of land use/cover changes

During Period I~IV, the land use/cover condition of our experimental site has

undergone two processes, one was the land degradation process (Period II~III), while
the other was the vegetation rehabilitation process (Period IV). Interesting phenomenon
was found that during these two processes: normalized $E_T$ values were both increased,
and normalized $E_T$ increased much faster from Period III to IV than that from Period I
to III.

The impact of phenological change on $E_T$ demonstrated that $E_T$ will decrease along

with the leaf browning. Thus, we expect that $E_T$ will also decease if only leaves were
cleared by human activities. However, during Period I~III, not only leaves were cleared,
but also all branches were cut-off, sand dunes (fixed and semi-fixed) were bulldozed,
the dry sand layers and the biological soil crusts (BSCs) were destroyed, making the
land use/cover condition complex. All these changed land surface properties might
contribute to the increase of $E_T$. The exists of dry sand layers and BSCs were
demonstrated to effectively restrained the soil evaporation rates (Wang et al., 2006; Lv





et al., 2006; Liu et al., 2006; Chen and Dong, 2001; Yang et al., 2015; Fu et al., 2010;
Liu, 2012). However, the bulldozing of sand dunes at our experimental site made the
elevation of the flat soil surface be lower than the average elevation of the undisturbed
soil surface (about 1.5 m, Figure 2(d)), which resulted that the groundwater depth was
much shallower than before. Thus, it is hard for the formation of dry sand layers with
shallower groundwater depth. In this situation with the destroyed BSCs and the
disappeared dry sand layers, the sufficient groundwater supply (Li and Li, 2000)
accelerated the loss of water that stored in shallow soil through evaporation. The
enhancement effect of soil evaporation offset the inhibition effect of transpiration by
leaves clearing, which made $E_T$ increase.

A secondary reason for the enhancement of soil evaporation was that more solar

radiation was absorbed by soil layer during land degradation process. In Period I with
natural vegetation, the radiation absorbed by shadowed soil was the solar radiation
transmitted into the canopy of shrub and grass. However, with the natural vegetation
being cut-off, the leaves and the branches were also removed, which made the
shadowed soil exposed and enhanced the radiation absorbed by soil, thus contribute to
the increase of soil evaporation (Martens et al., 2000; Panferov et al., 2001). Besides,
the removal of leaves and branches and the disappearance of sand dunes altered the
land surface albedos. Various scholars have demonstrated that changes of land surface
albedos could directly alter the solar radiation absorbed by the land surface (Dirmeyer
and Shukla, 1994; Greene et al., 1999), subsequently leading to the change in $E_T$.

Some inconsistent results were found from the previous analyses that aim at





studying the characteristics of $E_T$ with land degradation. For instance, Li et al. (2013)
have analyzed the features of $E_T$ during land degradation process in Qinghai-Tibet
Plateau, and they found that warming air temperature was the main cause to enhance
$E_T$. However, some other scholars have opposite conclusions. For example, Snyman
(2001) have compared $E_T$ of natural grassland and degraded grassland resulted from
overgrazing in a semi-arid are of South Africa, and he found that $E_T$ was smaller of the
degraded grassland due to less transpiration. Souza and Oyama (2011) have
demonstrated that desertification in a semi-arid area of Northeast Brazil contributed to
the decrease of $E_T$ due to the loss of transpiration from natural vegetation. Lu et al.
(2011) have found that $E_T$ was lower in disturbed grazed grassland compared to the
undisturbed grassland, and the lower soil water content was thought to be the
explanation to result in the decrease of $E_T$. Mao and Cherkauer (2009) have
demonstrated that $E_T$ decreased when land use/cover condition was changed from
forests to grass or cropland in the Great Lakes region. Furthermore, Hoshino et al. (2009)
have demonstrated that there was no difference in $E_T$ during the land degradation by
overgrazing in a semi-arid Mongolian grassland, and they thought that the reason might
be short time of grazing (2 years). Throughout the above researches of $E_T$ under land
degradation process, we found it was hard to accurately describe the features of $E_T$ even
when the land degradation was only performed by less vegetation coverage. Therefore,
in our study site with complex land surface properties (sand dunes, dry sand layers and
BSCs), the impact of land degradation on $E_T$ was much more complicated.
During the vegetation rehabilitation process (Period IV), normalized $E_T$ increased





significantly due to the rehabilitation of grass in zone B, even though with less
precipitation compared with other three periods. The rehabilitation of grass, rather than
shrub, was due to the sufficient groundwater supply resulted from bulldozing the sand
dunes. Previous researchers reported that sparse shrub more commonly grew at the top
of sand dunes and grass grew at the bottom of sand dunes, because the differences
between groundwater depth and the top of sand dunes was larger than that the that
between groundwater depth and the bottom of the sand dunes (Lv et al., 2006; Chen
and Dong, 2001). Because transpiration increases with the greening of vegetation (this
was demonstrated in section 3.3), the regrowing grass will enhance plant transpiration
supplied by the sufficient groundwater. More importantly, the transpiration rate of grass
is higher than that of shrubs because shrubs are more tolerant to drought (Yang et al.,
2014; Wang et al., 2002; Wu, 2006). Therefore, in the vegetation rehabilitation process,
the increasing rate of transpiration in Period IV was much higher than that in Period
I~III. Consistent conclusions of $E_T$ increase during vegetation rehabilitation process
were reported. For example, Qiu et al. (2011) have demonstrated that in the vegetation
rehabilitation process, $E_T$ increased and more water was consumed and less rainfall
would infiltrate deeper soil layer. Yang et al. (2014) and Sun et al (2006) also considered
$E_T$ would increase with vegetation rehabilitation due to the increase of transpiration.
Furthermore, Li et al. (2009) have reported that $E_T$ increased when land use/cover
condition converted from shrubland to grassland. Meanwhile, the regrowing grass can
reduce the radiation absorbed by soil and hence reduce the soil evaporation. However,
the intercept of radiation by grass canopy was expected to be smaller than that by mixed





shrub and grass canopy in Period I~III because the leaf area index of grass is smaller
than the sum of leaf area and stem area index of shrub. Therefore, the reduction of soil
evaporation in Period IV may be small compared with the increment of soil evaporation
in Period I~III.

We noticed that groundwater level decreased continuously from Period III due to

the enhancement of $E_T$ by the re-growth of grass and relative lower precipitation, and
the regrowing grass has a high transpiration rate compared with the native mixed shrub
and grass ecosystem. Therefore, we hypothesized that if the land use/cover condition
of zone B continues to be grassland in several years, the groundwater level will decrease
due to the larger consumption, making the soil water condition gradually become poor
for the growing of grass. Then, in this situation, the grassland is expected to degrade to
shrubland in zone B because shrubs are more tolerant to survive in water-starved
ecosystems. On the other hand, potato is studied to consume more than 320 mm in the
growing season (Qin et al., 2013; Liu et al., 2010) and the consumption is more than
that of natural grass (Qin et al., 2013, 2014; Hou et al., 2010). Besides, planting potato
needs to irrigate several times during the growing season (Fulton et al., 1970; Liu et al.,
2010; Fabeiro et al., 2001). As potato consumes much more water than, our result
implied that the groundwater level may continue to decrease faster with the growth of
potato, which may lead to a more fragile ecosystem.

5 Conclusion

In this study, seasonal and inter-annual features of $E_T$ were analyzed. The daily $E_T$



was in a range from 0.0 mm day$^{-1}$ to 6.8 mm day$^{-1}$ during the four periods. NDVI was
the predominant factor that influences the seasonal variations in $E_T$. Vegetation
greening had a positive effect on $E_T$. During land degradation process (Period I~Period
III), when natural vegetation (including leaves and branches), sand dunes, dry sand
layers and BSCs were all bulldozed by human activities, $E_T$ was observed to increase
with a mild rate. In vegetation rehabilitation process (Period IV) with sufficient
groundwater, $E_T$ also increased with a faster rate than that in the degradation process.
When land use/cover changed by human activities, the underlying mechanisms that
leads to the changes of $E_T$ were complex, and vegetation types, topography and soil
surface characteristics could contribute to the changes in $E_T$.

**Acknowledgements**
This research was supported by the National Key Research and Development Program
(2016YFC0402404) and the Basic Research Fund Program of State key Laboratory of
Hydroscience and Engineering (Grant No. 2014-KY-04). We thank A. W. Jayawardena
for language and constructive suggestions of the manuscript.

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

Chinese).





**Figure and table captions**
Fig. 1. Location of the Loess Plateau and the map of study site (LP: the Loess Plateau;
black triangle: flux tower; white triangle: Yulin meteorological station; ①: Tu River;
②: Yuxi River; ③: Yellow River).

Fig. 2. Land use/cover conditions of the study site over the Loess Plateau: (a) the natural
land use/cover condition of shrubland (photo was taken at 6th August 2011); (b) the
natural land use/cover condition of grassland (photo was taken at 7th September 2011);
(c) the undisturbed zone (natural vegetation) and the disturbed zone (bare soil) in the
land degradation process (photo was taken at 26th April 2013); (d) the undisturbed zone
(natural vegetation) and the disturbed zone (grassland) in the vegetation rehabilitation
process (photo was taken at 16th August 2014).

Fig. 3. Diagrams of wind rose and footprint (a) wind rose of study site by using half-
hourly wind speed and wind direction data; (b) simulated footprint by ellipse (the long
axis is 1682m, and the short axis is 1263m; zone A is the source area that have not
encountered any land use/cover change, while zone B is the source area that have
experienced land use/cover change by human activities; white triangle is the flux tower).

Fig. 4. Seasonal characteristics of monthly (a) sunshine duration ($D_S$); (b) temperature
($T_a$); (c) relative humidity ($R_H$); (d) total precipitation ($P$) of four periods at the study
site and climatological normal (1954-2014 climatological normal in Yulin
meteorological station).




Fig. 5. Seasonal and inter-annual characteristics of daily (a) evapotranspiration ($E_T$,
mm); (b) potential evapotranspiration ($E_{TP}$, mm); (c) NDVI in zone A and zone B within
the footprint; (d) precipitation ($P$, mm); (e) soil water stress of root zone ($f_s$) during 1st
July 2011 to 30th June 2015.

Fig. 6. The correlations between daily evapotranspiration ($E_T$, mm) and its controlling
factors: (a) daily potential evapotranspiration ($E_{TP}$, mm); (b) daily weight-averaged
NDVI within footprint ($NDVI_w$); (c) daily soil water stress of root zone ($f_s$) in Period I
excluding the data in rainy days (r: Pearson's correlation significance; T: T-test
significance).

Fig. 7. Quantitative analysis between (a) vegetation phenological change ($NDVI_w$) and
daily normalized $E_T$ ($f_v = E_T/(E_{TP} \times f_s)$) in Period I (exclude the data in rainy days
and frozen days); (b) the indicator of land use/cover change ($D_{lu}$) and total normalized
$E_T$ ($f_v = E_T/(E_{TP} \times f_s)$) of the growing season in each period.

Table 1. Daily air temperature ($Ta$, ℃), relatively humidity ($R_H$, %), sunshine duration
($Ds$, h), soil water content of the root zone ($\theta_r$, m³ m⁻³), the groundwater level (GWL,
m) and precipitation ($P$, mm) in 1954-2014 and in the growing season of each period
(Because there were some missing data in Period IV (from 12th September 2014 to 23th
November 2014 and from 13th March 2015 to 22th April 2015), we got rid of data in





these two time range of Period I~III and 1954-2014)

| Items | 1954-2014 | I | II | III | IV |
|---|---|---|---|---|---|
| $T$a (℃) | 19.8 | 19.6 | 20.4 | 19.9 | 19.3 |
| $R_H$ (%) | 57.7 | 57.3 | 54.9 | 53.4 | 52 |
| $D$s (h) | 213.3 | 220.7 | 215.8 | 218.2 | 220.7 |
| P (mm) | 329.8 | 357.1 | 384.1 | 330.2 | 199.8 |
| $\theta_r$ (m$^3$ m$^{-3}$) | _ | 0.077 | 0.077 | 0.076 | 0.064 |
| GWL (m) | _ | -3.8 | -3.6 | -3.0 | -3.5 |


Table 2. Typical values of total evapotranspiration ($E_T$, mm), total potential
evapotranspiration ($E_{TP}$, mm), indicator of land use/cover change ($D_{lu}$), soil water stress
of root zone ($f_s$) and normalized $E_T$ (i.e., $f_v$ $(= E_T/(E_{TP} \times f_s))$) in the growing season
of each period. (Because there were some missing data in Period IV (from 12[th]
September 2014 to 23[th] November 2014 and from 13[th] March 2015 to 22[th] April 2015),
we removed the values of $E_T$, $E_{TP}$ and $f_s$ of these two time ranges in Period I~III).

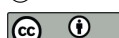


| | Items | $E_T$ | $E_{TP}$ | $D_{lu}$ | $f_s$ | $f_v$ |
|---|---|---|---|---|---|---|
| | Periods | (mm) | (mm) | (%) | (dimensionless) | (dimensionless) |
| | I | 238.4 | 876.1 | -0.2 | 0.62 | 78.1 |
| Growing | II | 236.5 | 870.7 | 4.6 | 0.63 | 79.9 |
| season | III | 292.1 | 956 | 10.4 | 0.59 | 86.3 |
| | IV | 332.2 | 937 | 6 | 0.37 | 111.9 |




Fig.1

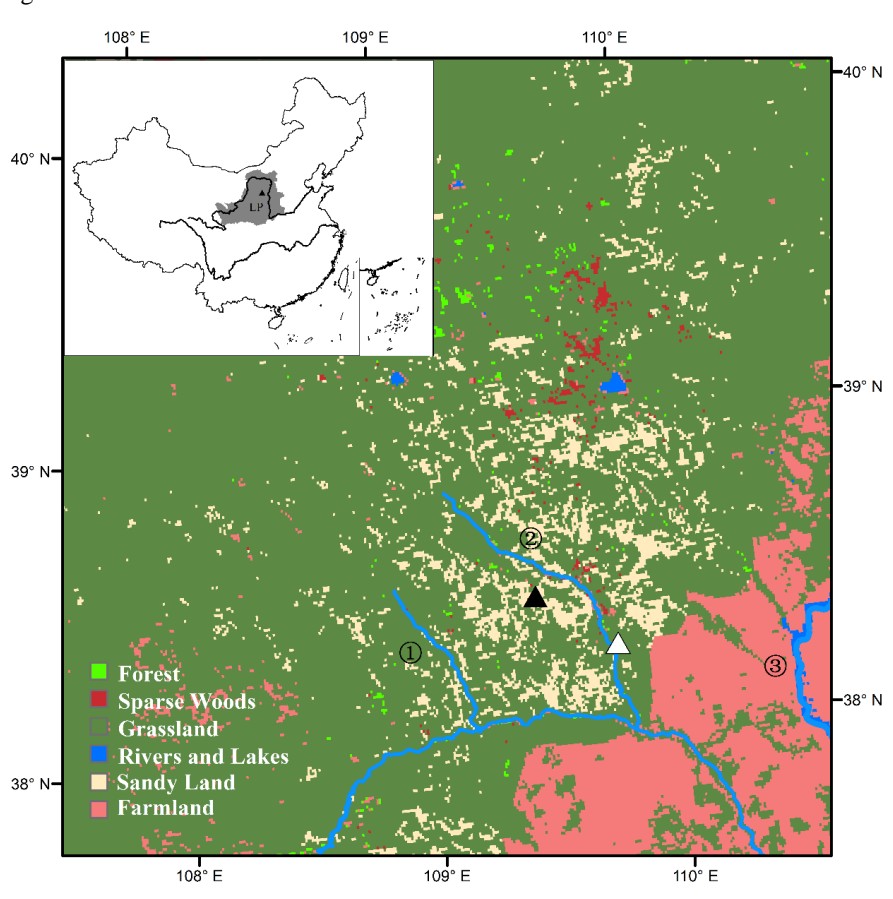




Fig.2

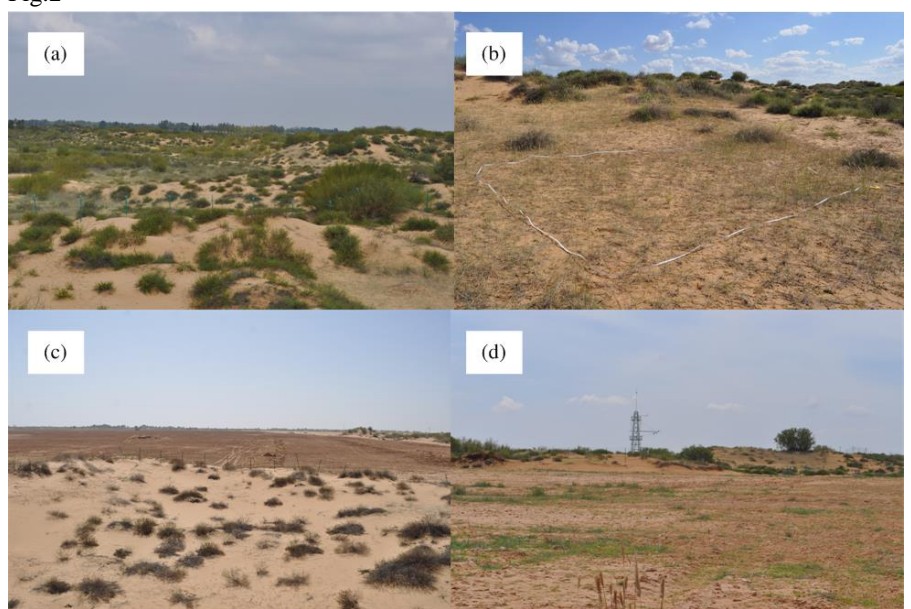


Fig.3

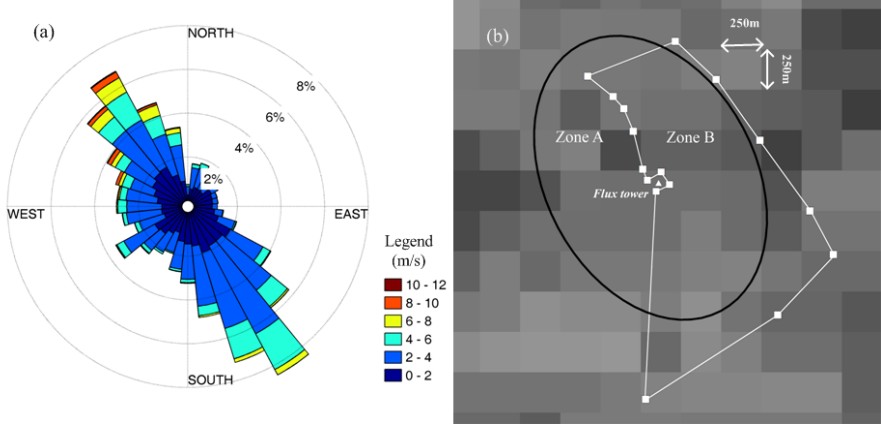






Fig.4

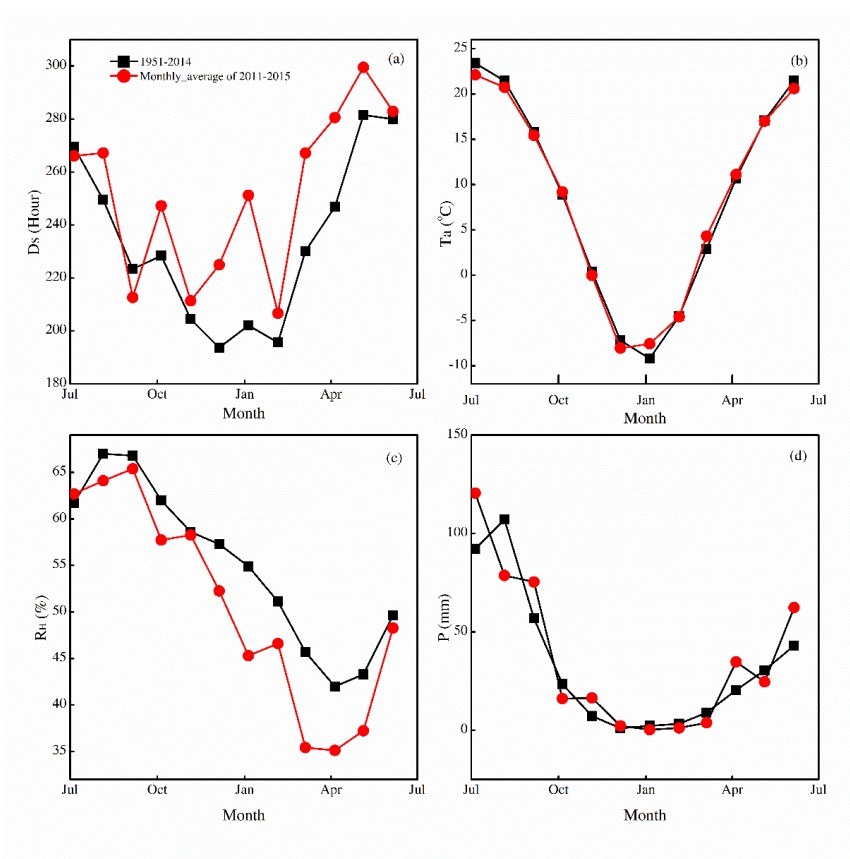





Fig.5

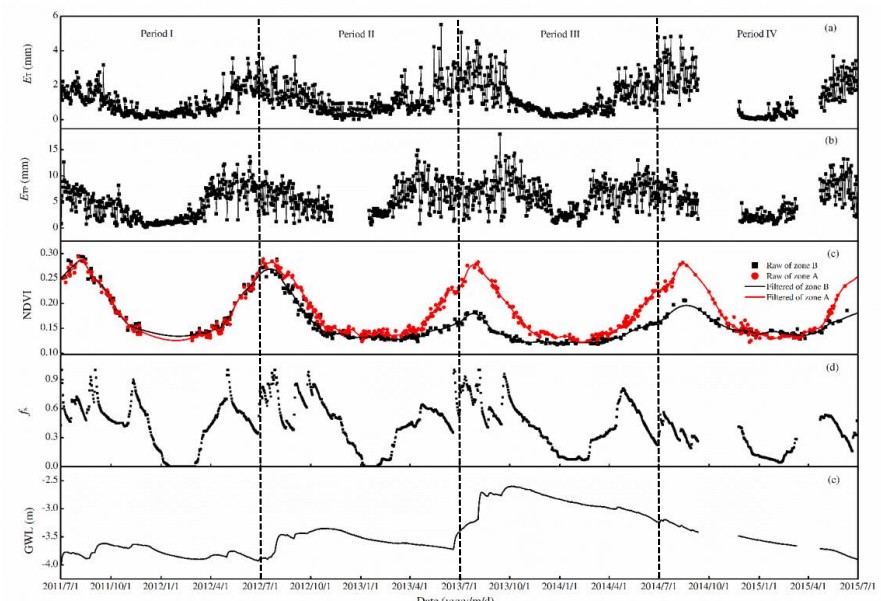


Fig.6

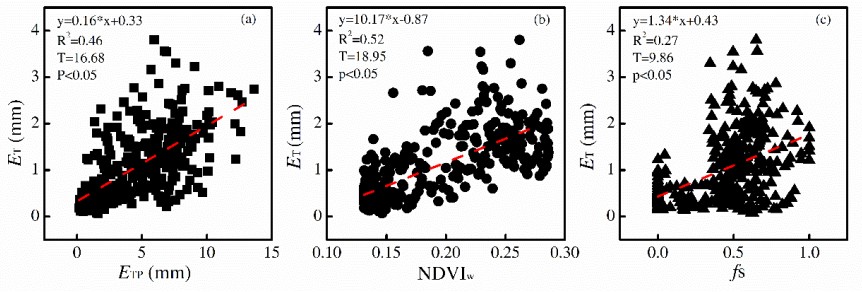




Fig.7

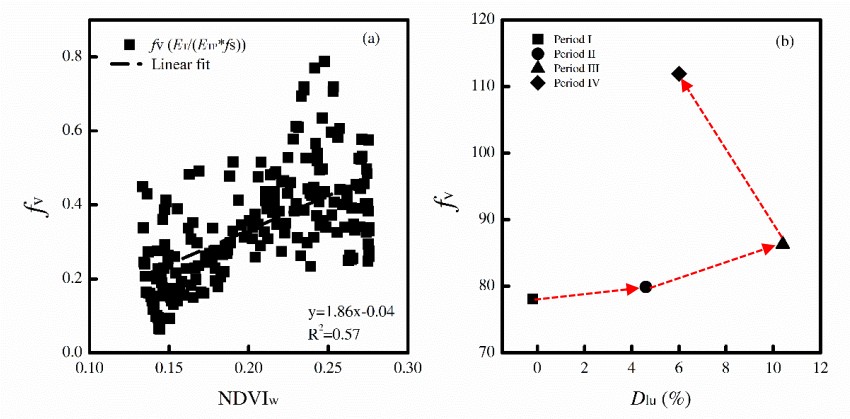


