# Peer review of "Manuscript Title"

_Hydrology and Earth System Sciences, 2016_

## Referee Comment (RC1) · Anonymous Referee #1 · 17 Oct 2016

It seems Gong and his/her colleagues have substantially improved the manuscript in the resubmitted version. Generally, the manuscript is written clearly and understand-able, but some grammars are still need to be checked and confirmed, probably by a native speaker.

I like the discussion about the human impact on evaporation, i.e. vegetation degra-dation and sand dunes bulldozing. The impact of vegetation degradation did not only change the vegetation cover but also modify the soil conditions. I agree with the authors that the processes are complex, and still needs to be further investigated. Summarily, these relative long-term and intensive land surface water and energy observations are important for us to understand the interaction between land surface and atmosphere and even groundwater, especially in this semiarid region where the ecosystems are vulnerable. But there is still space to improve the quality of this manuscript before

publication.

Other comments:

L24: I think it might be okay to generalize the results a little bit to "improve our understanding . . . in the fragile ecosystems of semiarid regions."

L34: Not clear info. Please rephrase.

L 65: limiting factor for. . .

L66: what do you mean by "common droughts"?

L 62-73: Some detailed information might better go to study site section.

L 81: in situ field. . . ?

L88: "doubtful" is a strong word. You'd better change it.

L97: ". . .is little learned. . ." reads awkward. Please rephrase. Again, "field observations"

L102 probably change measurements to measurement

L123: is it better to say "water demand"?

L141: "as time went on. . .". Please keep the same tense in one sentence.

L187-189: It might be better to briefly describe how you calculated latent heat flux.

L198: what do you mean by "immediately"?

L266: How did you determine the factors in this equation?

L291-294: Is this the commonly used method to calculate NDVI? If so, you do not need to mention these details. And I have no idea why you describe the NDVI_Terra and NDVI_Aqua. Can you clarify?

L398-399: Since NDVI is a normalized factor and you derived the NDVI_w based NDVI,

I do not think it is meaningful to quantify the impact of NDVI on evaporation. This relationship might be changed in different cases and even with different time series datasets. You can describe this relation, but it is probably not suitable as a highlight and mention it in Abstract.

L412: do you mean "compared Period I with..."?

L431-434: The first-order control of evaporation is a long time debate. I agree with the conclusion, but this research might be not directly related to this conclusion. I suggest the authors weaken the tone, to use "probably" or "very likely" etc.

L545: "tolerant to" is probably followed by some "vices", not survive. Please rephrase.

L 550: more water than "what"?

---

## Referee Comment (RC2) · D. Guo (Referee) · 29 Oct 2016

Review HESS-2016-490 Overview This study assessed the relationships between evapotranspiration (ET) and change of land by analyzing the eddy-covariance measurements of actual ET together with data of a number of its potentially influencing factors including normalized vegetation index, soil water content as well as climate variables to estimate potential ET. Data are collected from a case study with different periods reflecting changes of land-use conditions, which provides further evidence to support the statistical analyses. The manuscript is well written and the knowledge promoted is a clear contribution to the understanding of how ET processes can potentially change with land-use in semiarid regions. I think this study is suitable for publication after moderate revision, with improved clarity and better flow especially for the Introduction and Methodology - please see my major comments below.

[Figure]

Major comments: 1. The Introduction launched quite well with highlighting the importance of the assessing relationship between ET to vegetation conditions in arid/semiarid regions (L28-37), followed by a comprehensive literature review explaining relevant physical mechanisms (L38-61). However, the third paragraph (L62-79) seems to be a bit disjointed as the flow of ET/vegetation stops and shifts to the case study, whereas paragraph 4 (L80-98) returns to the ET/vegetation flow and paragraph 5 again introduces the study site. I think the easiest way to improve the flow is by swapping paragraph 3 and 4 (I found this can in fact fit better with your current connecting sentences between paragraphs i.e. L60-61, L96-98). So you would have:

Paragraph 1: importance of the assessing relationship between ET to vegetation conditions in arid/semiarid regions Paragraph 2: physical mechanisms on how vegetation can influence ET (finish with L60-61 which then leads to the method of assessing these impacts) Paragraph 3: method to assess the vegetation impact on ET (finish with L96-98 which then leads to the case study in a sparse shrubland) Paragraph 4: introducing the case study and how it can contribute to the above-mentioned knowledge gap. I'd also recommend combining this use some discussion from the current paragraph 5 (L99-101) to help justifying the choice of the case study. Paragraph 5: I'd recommend to leave this paragraph purely as a summary of the study (as current L102-104), and maybe elaborate a little bit with highlighting the significance of the study.

I think the above structure can allow the storyline about ET/vegetation relationship to complete before introducing the study site, which provides a smoother transition and also better justification on the use of Mu US sandland as the case study.

2. I appreciate the comprehensiveness of Section 2 which covers the details of data collections methods and models used to analyze different data variables. However, I found that Section 2.3.2 become a bit confusing with introducing models related to a number of variables. As this section describes the methods employed for the core analyses of the study, I think the clarity can be further improved by using further subsections for individual variables. In addition, I think the methods used for data analyses

should be introduced as well. Currently the statistical methods used for data analyses are mainly described in the Results section (e.g. L325-327, L380-384, L386-L389). I think it can be clearer to summarize them in the Section 2.3.2 instead (probably as an overview in the start of this section). In this way you can better justify why these analyses are conducted and how they help to answer the research questions, while purely focusing on the results and interpretation in the Results section. And then the readers can get an overall understanding on the data analyses to be conducted and knowing what to expect in the Results section.

So I'd suggest the following structure for Section 2.3.2: Sub-section 1: overview - introducing the variables which are needed for analyzing the impact of ET and vegetation conditions (these will be detailed in the following sub-sections), and what analyses will be conducted with these variables (e.g. as those introduced in L325-327, L380-384 and L386-L389 etc.) Sub-section 2: estimating potential ET Sub-section 3: estimating soil water content Sub-section 4: estimating NDVI ...

3. I think the Section 2 (Material and Methods) is a bit too long trying to cover different aspects including case study, measurements of raw data, data processing and analyzing. In my opinion a better way to organize these is to break Section 2 into two sections, for example as: Section 2. Case study and data (note:I'd use 'data' to refer to the raw measurements here rather than in the next section, where you introduce data-processing and analyzing.) 2.1 site description 2.2 measurements ... Section 3. Methodology 3.1 flux data processing 3.2 footprint model 3.3 method of analyzing controlling factors of ET (and if you agree with my last comment, the sub-sections can go below:) 3.3.1 ... 3.3.2 ... ...

Minor comments: 1. L30: 'ET' - please define acronym when it first appears in the text, and please also check if all other acronyms are properly defined. 2. L101: '4' - please spell out numbers less than 10 i.e. as 'four-year'. 3. L111: please delete the repeated 'temperate'. Also, is there a better way to introduce the climate zone, as currently it seems like a 'noun train' ('temperate semiarid continental monsoon climate'). You can find some examples on improving 'noun train' from http://www.webwritingthatworks.com/DGuideCOG5b.htm. 4. L194-195: Would there be any impact on the results from this data removal, and would this be a limitation of the study? This should be briefly discussed (Maybe in the Discussion or Conclusion section?). 5. L208-211: It would be clearer if these lines can be presented as individual formulae (i.e. in the format of L219). Also, according to L205, the 'n' in 'Rn' should be subscripted - please also check that the use of other symbols is consistent throughout the text. 6. L246: 'psychrometric constant' - what is the value of the constant? 7. L248: 'U2' - where is it in Equation (5)? 8. L337: 'Ds' - not defined as in Minor comment #1. Also, how are the data of Ds obtained? I couldn't seem to find it in Section 2.2.2 (other measurements). 9. L337: 'normal' - I think 'average monthly' would be a better description here. 10. L347: Figure 4 has not been introduced in the text yet, should it be mentioned somewhere between L336-337? 11. L380: 'relationships' - 'correlations' would be a more accurate description. 12. L389-390: the r2 only investigates linear relationships - are you expecting any non-linear relationships which are not covered here and would this be a limitation? This can be briefly discussed. 13. L464: The term 'BSC' has already been defined in L68. 14. L565: It should be worth highlighting some significance and contributions of this study towards the end of conclusion. 15. Fig. 6: I don't think the use of different shapes is necessary given that you are using multiple panels? 16. L884 (title of Fig. 6): 'r: Pearson's correlation significance' should r be 'Pearson's correlation coefficient' instead?

---

## Author Comment (AC1) · 19 Dec 2016

**Reply to Referee #1**

It seems Gong and his/her colleagues have substantially improved the manuscript in the resubmitted version. Generally, the manuscript is written clearly and understandable, but some grammars are still need to be checked and confirmed, probably by a native speaker. I like the discussion about the human impact on evaporation, i.e. vegetation degradation and sand dunes bulldozing. The impact of vegetation degradation did not only change the vegetation cover but also modify the soil conditions. I agree with the authors that the processes are complex, and still needs to be further investigated. Summarily, these relative long-term and intensive land surface water and energy observations are important for us to understand the interaction between land surface and atmosphere and even groundwater, especially in this semiarid region where the ecosystems are vulnerable. But there is still space to improve the quality of this manuscript before publication.

Answer: Thank you for your positive comments. We will check and confirm the grammars of the manuscript by a native speaker.

Other comments:

L24: I think it might be okay to generalize the results a little bit to "improve our understanding . . . in the fragile ecosystems of semiarid regions."

Answer: We will emphasize the significance of this study in the abstract with the following sentences:

"This study improves our understanding of land use/cover change impact on evapotranspiration and provides a scientific reference to the regional land management in the context of water resources sustainability".

L34: Not clear info. Please rephrase.

Answer: We will rephrase it with the following sentences.

"In terms of physical processes, ET is affected by net radiation (Valipour et al., 2015), water vapor pressure deficit (Zhang et al., 2014), wind speed (Falamarzi et al., 2014), and soil moisture stress (Allen et al., 1998). Besides, vegetation is also a crucial factor influencing ET (Tian et al., 2015; Wang et al., 2011; Piao et al., 2006; Mackay et al., 2007)".

L 65: limiting factor for. . .

Answer: We will replace the expression of "limiting factor on vegetation" with "limiting factor for vegetation".

L66: what do you mean by "common droughts"?

Answer: In this sentence, the "common droughts" referred to droughts. In order to avoid misunderstanding, we will delete "common" in this sentence.

L 62-73: Some detailed information might better go to study site section.

Answer: We used the information in this paragraph to describe the typical characteristics of Mu Us sandland, including the sand dunes, biological soil crusts

(BSCs) and dry sand layer, which result in complex ET process. Therefore, we think it will be better to leave some information about these typical land surface properties in this paragraph. Following this suggestion, we will move some sentences to the section of site description.

L 81: in situ field. . . ?
Answer: We will correct it.

L88: "doubtful" is a strong word. You'd better change it.
Answer: We will revise it by replacing "doubtful" with "may induce uncertainty".

L97: ". . .is little learned. . ." reads awkward. Please rephrase. Again, "field observations"
Answer: We will correct grammar and spell mistake.

L102 probably change measurements to measurement
Answer: We will revise it.

L123: is it better to say "water demand"?
Answer: Yes, we will revise it.

L141: "as time went on. . .". Please keep the same tense in one sentence.
Answer: We will revise it.

L187-189: It might be better to briefly describe how you calculated latent heat flux.
Answer: Thank you for your suggestion. We will add a brief description of latent heat flux calculations.

L198: what do you mean by "immediately"?
Answer: We used "immediately" to emphasize that we used the values before and after the data gap. In order to avoid the confusion, we will delete the word "immediately".

L266: How did you determine the factors in this equation?
Answer: We will add the detailed descriptions in this section.

L291-294: Is this the commonly used method to calculate NDVI? If so, you do not need to mention these details. And I have no idea why you describe the NDVI_Terra and NDVI_Aqua. Can you clarify?
Answer: Yes, this method is commonly used to calculate NDVI. As we found that there were tiny differences ( $\left| NDVI_{Terra} - NDVI_{Aqua} \right| = 0.01 \pm 0.0075$ ) between the calculated daily $NDVI_{Terra}$ and $NDVI_{Aqua}$ , we calculated NDVI by averaging $NDVI_{Terra}$ and $NDVI_{Aqua}$ in our study in order to eliminate the impacts caused by such difference.

We will add the above information in the text. Besides, we will follow your

suggestion to simplify the descriptions of the method to calculate NDVI.

L398-399: Since NDVI is a normalized factor and you derived the NDVI_w based NDVI, I do not think it is meaningful to quantify the impact of NDVI on evaporation. This relationship might be changed in different cases and even with different time series datasets. You can describe this relation, but it is probably not suitable as a highlight and mention it in Abstract.

Answer: We agree with you that the relationship between NDVI and ET differs in different cases or with different time series. We also discussed this point in the manuscript. The main purpose for describing the relationship is to compare our result with other studies in different cases to show how strong NDVI affects ET. By our survey, the relationship between NDVI and ET were reported mostly in forests (Loukas et al., 2005; Nouri et al., 2014; Chong et al., 2007) and grassland (Kondoh and Higuchi, 2001; Nouri et al., 2014). Thus, it is meaningful to fill the gap to quantify the impact of NDVI on ET for the shrubland.

In summary, we described this relationship in the main body for comparison, and following this suggestion, we will delete this sentence from the abstract.

L412: do you mean "compared Period I with. . ."?

Answer: Yes, we will revise the sentence.

L431-434: The first-order control of evaporation is a long time debate. I agree with the conclusion, but this research might be not directly related to this conclusion. I suggest the authors weaken the tone, to use "probably" or "very likely" etc.

Answer: We will revise the word "mainly" in this sentence to "probably".

L545: "tolerant to" is probably followed by some "vices", not survive. Please rephrase.

Answer: We will revise the sentence.

L 550: more water than "what"?

Answer: The missing "what" in this sentence was the word "grass". However, we will delete this sentence "As potato consumes much more water than grass". Because we have emphasized the fact that potato consumes more water than grass in this paragraph.

---

## Author Comment (AC2) · 19 Dec 2016

Reply to Referee 2#

**Overview** This study assessed the relationships between evapotranspiration (ET) and change of land by analyzing the eddy-covariance measurements of actual ET together with data of a number of its potentially influencing factors including normalized vegetation index, soil water content as well as climate variables to estimate potential ET. Data are collected from a case study with different periods reflecting changes of land-use conditions, which provides further evidence to support the statistical analyses. The manuscript is well written and the knowledge promoted is a clear contribution to the understanding of how ET processes can potentially change with land-use in semiarid regions. I think this study is suitable for publication after moderate revision, with improved clarity and better flow especially for the Introduction and Methodology - please see my major comments below.

Answer: Thank you for your positive comments. We will revise our manuscript by carefully following your comments and suggestions.

**Major comments:**
**1.** The Introduction launched quite well with highlighting the importance of the assessing relationship between ET to vegetation conditions in arid/semiarid regions (L28-37), followed by a comprehensive literature review explaining relevant physical mechanisms (L38-61). However, the third paragraph (L62-79) seems to be a bit disjointed as the flow of ET/vegetation stops and shifts to the case study, whereas paragraph 4 (L80-98) returns to the ET/vegetation flow and paragraph 5 again introduces the study site. I think the easiest way to improve the flow is by swapping paragraph 3 and 4 (I found this can in fact fit better with your current connecting sentences between paragraphs i.e. L60-61, L96-98). So you would have: Paragraph 1: importance of the assessing relationship between ET to vegetation conditions in arid/semiarid regions Paragraph 2: physical mechanisms on how vegetation can influence ET (finish with L60-61 which then leads to the method of assessing these impacts) Paragraph 3: method to assess the vegetation impact on ET (finish with L96-98 which then leads to the case study in a sparse shrubland) Paragraph 4: introducing the case study and how it can contribute to the above-mentioned knowledge gap. I'd also recommend combining this use some discussion from the current paragraph 5 (L99-101) to help justifying the choice of the case study. Paragraph 5: I'd recommend to leave this paragraph purely as a summary of the study (as current L102-104), and maybe elaborate a little bit with highlighting the significance of the study. I think the above structure can allow the storyline about ET/vegetation relationship to complete before introducing the study site, which provides a smoother transition and also better justification on the use of Mu US sandland as the case study.

Answer: We will re-arrange these paragraphs by following your suggestions.

**2.** I appreciate the comprehensiveness of Section 2 which covers the details of data collections methods and models used to analyze different data variables. However, I found that Section 2.3.2 become a bit confusing with introducing models related to a number of variables. As this section describes the methods employed for the core

analyses of the study, I think the clarity can be further improved by using further subsections for individual variables. In addition, I think the methods used for data analyses should be introduced as well. Currently the statistical methods used for data analyses are mainly described in the Results section (e.g. L325-327, L380-384, L386-L389). I think it can be clearer to summarize them in the Section 2.3.2 instead (probably as an overview in the start of this section). In this way you can better justify why these analyses are conducted and how they help to answer the research questions, while purely focusing on the results and interpretation in the Results section. And then the readers can get an overall understanding on the data analyses to be conducted and knowing what to expect in the Results section. So I'd suggest the following structure for Section 2.3.2: Sub-section 1: overview – introducing the variables which are needed for analyzing the impact of ET and vegetation conditions (these will be detailed in the following sub-sections), and what analyses will be conducted with these variables (e.g. as those introduced in L325-327, L380-384 and L386-L389 etc.) Sub-section 2: estimating potential ET Sub-section 3: estimating soil water content Sub-section 4: estimating NDVI ...

Answer: We will re-arrange the structure of this section by following your suggestion.

**3.** I think the Section 2 (Material and Methods) is a bit too long trying to cover different aspects including case study, measurements of raw data, data processing and analyzing. In my opinion a better way to organize these is to break Section 2 into two sections, for example as: Section 2. Case study and data (note:I'd use 'data' to refer to the raw measurements here rather than in the next section, where you introduce data-processing and analyzing.) 2.1 site description 2.2 measurements ... Section 3. Methodology 3.1 flux data processing 3.2 footprint model 3.3 method of analyzing controlling factors of ET (and if you agree with my last comment, the sub-sections can go below:) 3.3.1 ... 3.3.2 ... ...

Answer: We will rearrange Section 2 and 3 by following your suggestion.

**Minor comments:**

**1.** L30: 'ET' - please define acronym when it first appears in the text, and please also check if all other acronyms are properly defined.

Answer: We will add the definition of ET.

**2.** L101: '4' - please spell out numbers less than 10 i.e. as 'four-year'.

Answer: We will revise it.

**3.** L111: please delete the repeated 'temperate'. Also, is there a better way to introduce the climate zone, as currently it seems like a 'noun train' ('temperate semiarid continental monsoon climate'). You can find some examples on improving 'noun train' from http://www.webwritingthatworks.com/DGuideCOG5b.htm.

Answer: We will revise it.

**4.** L194-195: Would there be any impact on the results from this data removal, and

would this be a limitation of the study? This should be briefly discussed (Maybe in the Discussion or Conclusion section?).

Answer: We will add necessary information about the missing data and discuss the limitation.

**5.** L208-211: It would be clearer if these lines can be presented as individual formulae (i.e. in the format of L219). Also, according to L205, the 'n' in 'Rn' should be subscripted - please also check that the use of other symbols is consistent throughout the text.

Answer: After considering your comment, we think it will be better to change these lines from several formulae to the following form: "$= a * (R_n - G)^2 + b * (R_n - G) + c$ (Period I: a = 0.0014, b = 0.075, c = 10.69, R = 0.77; Period II: a = 0.0012, b = 0.056, c = 17.69, R = 0.67; Period III: a = 0.0014, b = 0.16, c = 13.24, R = 0.75; Period IV: a = 0.0015, b = -0.083, c = 25.87, R = 0.69)"

In addition, we will revise the $R_n$ and check the use of other symbols throughout the text.

**6.** L246: 'psychrometric constant' - what is the value of the constant?

Answer: We will add the equation of psychrometric constant in the text.

**7.** L248: 'U2' - where is it in Equation (5)?

Answer: $U_2$ is used to calculate the aerodynamic resistance ($r_a$). We will move the equation of calculating $U_2$ from Eq.6 to Eq.8 in the text.

**8.** L337: 'Ds' - not defined as in Minor comment #1. Also, how are the data of Ds obtained? I couldn't seem to find it in Section 2.2.2 (other measurements).

Answer: Thank you for your kind remind. We will add the information of measurement that obtained $D_s$ in the section of "other measurements".

**9.** L337: 'normal' - I think 'average monthly' would be a better description here.

Answer: According to your comment, we plan to revise "normal" to "long-term average monthly values".

**10.** L347: Figure 4 has not been introduced in the text yet, should it be mentioned somewhere between L336-337?

Answer: Yes, we will add the sentence of "Four-year and long-term (1954-2014) average monthly values of $D_s$, $T_a$, $R_H$, P were showed in Fig.4." in this section.

**11.** L380: 'relationships' - 'correlations' would be a more accurate description.

Answer: Thank you for your suggestion. We will revise the word "relationships" to "correlations".

**12.** L389-390: the r2 only investigates linear relationships - are you expecting any non-linear relationships which are not covered here and would this be a limitation? This can

be briefly discussed.

Answer: In fact, we have used several common functions (e.g., exponential function, linear function, logarithmic function and quadratic function) to fit the correlations between ET and its controlling factors ($E_{TP}$, NDVI and $f_s$). The values of determination coefficient ($R^2$) are listed in the following Tab.1.

According to the results shown in Table 1, value of $R^2$ for the linear function is the highest. Therefore, we chose the linear function.

We will add the above information in the section of "Statistical analysis".

Table 1. The determination coefficient ($R^2$) of the correlations between ET and the three controlling factors.

|  | ET and $E_{TP}$ | ET and NDVI | ET and $f_s$ |
| --- | --- | --- | --- |
| Exponential function | 0.46 | 0.54 | 0.27 |
| Linear function | 0.46 | 0.54 | 0.28 |
| Logarithmic function | 0.43 | 0.53 | 0.19 |
| Quadratic function | 0.45 | 0.54 | 0.28 |

**13.** L464: The term 'BSC' has already been defined in L68.

Answer: We will delete the definition here.

**14.** L565: It should be worth highlighting some significance and contributions of this study towards the end of conclusion.

Answer: We will highlight significance and contributions of this study at the end of conclusion.

**15.** Fig. 6: I don't think the use of different shapes is necessary given that you are using multiple panels?

Answer: We will revise Fig.6.

**16.** L884 (title of Fig. 6): 'r: Pearson's correlation significance' should r be 'Pearson's correlation coefficient' instead?

Answer: Yes, we will revise it in the title of Fig.6.

---

## Author Response (AR1)

**Reply to Referee #1**

It seems Gong and his/her colleagues have substantially improved the manuscript in the resubmitted version. Generally, the manuscript is written clearly and understandable, but some grammars are still need to be checked and confirmed, probably by a native speaker. I like the discussion about the human impact on evaporation, i.e. vegetation degradation and sand dunes bulldozing. The impact of vegetation degradation did not only change the vegetation cover but also modify the soil conditions. I agree with the authors that the processes are complex, and still needs to be further investigated. Summarily, these relative long-term and intensive land surface water and energy observations are important for us to understand the interaction between land surface and atmosphere and even groundwater, especially in this semiarid region where the ecosystems are vulnerable. But there is still space to improve the quality of this manuscript before publication.

Answer: Thank you for your positive comments. The manuscript has been checked and revised by the English-speaking editors of AJE (the supplement is the certificate).

Other comments:

L24: I think it might be okay to generalize the results a little bit to "improve our understanding . . . in the fragile ecosystems of semiarid regions."

Answer: We changed the significance of this study in the abstract with the following sentences, please see lines 18-20 in the revised manuscript.

"This study could improve our understanding of the effects of land use/cover change on ET in the fragile ecosystem of semiarid regions and provide a scientific reference for the sustainable management of regional land and water resources"

L34: Not clear info. Please rephrase.

Answer: We rephrased this sentence with the following sentences.

"In terms of physical processes, ET is affected by net radiation (Valipour et al., 2015), water vapor pressure deficit (Zhang et al., 2014), wind speed (Falamarzi et al., 2014), and soil water stress (Allen et al., 1998). Moreover, vegetation condition is also a crucial factor influencing ET (Tian et al., 2015; Wang et al., 2011; Piao et al., 2006; Mackay et al., 2007)", please see lines 29-33 in the revised manuscript.

L 65: limiting factor for. . .

Answer: We replaced the expression of "limiting factor on vegetation" with "limiting factor for vegetation", please see line 115 in the revised manuscript.

L66: what do you mean by "common droughts"?

Answer: In this sentence, the "common droughts" referred to droughts. In order to avoid misunderstanding, we deleted "common" in this sentence, please see line 116.

L 62-73: Some detailed information might better go to study site section.

Answer: We used the information in this paragraph to describe the typical characteristics of Mu Us Sandy Land, including the sand dunes, biological soil crusts (BSCs) and dry sand layer, which result in complex ET process. Therefore, we thought it might be better to leave some information about these typical land surface properties in this paragraph. And following your suggestion, we moved the following sentences describing the vegetation and water condition of Mu Us Sandy Land to the section 2.1 (Site description), please see lines 115-117 in the revised manuscript.

"Shortage of water is the critical limiting factor for vegetation in this regions, and drought-enduring vegetation are prevailed as a result of droughts (Wang et al., 2002; Wu, 2006). There are at least 117 shrub and semi-shrub species within the Mu Us Sandland (Dong and Zhang, 2001)."

L 81: in situ field. . . ?
Answer: Yes, we corrected it.

L88: "doubtful" is a strong word. You'd better change it.
Answer: Following your suggestion, we revised it by replacing "doubtful" with "may induce uncertainty", please see line 65 in the revised manuscript.

L97: ". . .is little learned. . ." reads awkward. Please rephrase. Again, "field observations"
Answer: We revised the sentence from "To our knowledge, there is little learned of ET under native sparse shrubland and continuous field observations under land degradation and vegetation rehabilitation conditions." to "Continuous field observations under both land degradation and vegetation rehabilitation processes have rarely been documented, especially in the semiarid shrubland", please see lines 74-76 in the revised manuscript.
   Yes, we corrected the word "field".

L102 probably change measurements to measurement
Answer: We revised the word "measurements" to "measurement", please see line 93 in the revised manuscript.

L123: is it better to say "water demand"?
Answer: Yes, we revised it.

L141: "as time went on. . .". Please keep the same tense in one sentence.
Answer: We revised it.

L187-189: It might be better to briefly describe how you calculated latent heat flux.
Answer: We added a brief description of latent heat flux calculations by eddypro in the section 2.3 (flux data processing) with the following sentences, please see lines 179-185 in the revised manuscript.

"10 Hz 3-dimensional wind speed and water vapor concentrations that collected by EC technique were processed to half-hourly latent heat flux ( $\lambda E_T$ ) using Eddypro processing software (v5.2.0, LI-COR, Lincoln, NE USA). The main principle is that $\lambda E_T$ can be expressed as $\rho_a \overline{w' q'}$ (where $w'$ is the fluctuation of vertical wind speed, $q'$ is the fluctuation of specific humidity and $\rho_a$ is the air density). The software also applies the quality control of data, including spike removal, tilt correction, time lag compensation, turbulent fluctuation blocking and spectral corrections."

L198: what do you mean by "immediately"?
Answer: We used "immediately" to emphasize that we used the values before and after the data gap. In order to avoid the confusion, we deleted the word "immediately" here, please see line 201 in the revised manuscript.

L266: How did you determine the factors in this equation?
Answer: In our study, $\theta_r$ was calculated by $\theta$ at different depths ( $\theta_i, i = 5, 10, 20, 40, 60, 80, 120, 160$ cm). A schematic diagram is showed in the following.

Surface

[Figure]

Figure 1. The schematic diagram of root-zone soil water content calculation

In the layer I (0-5 cm), the soil water profile was assumed triangular, while in the layers II, III, IV, V, VI, VII and VIII, the soil water profiles were assumed trapezoidal. Therefore, $\theta_r$ was calculated by the following equation,

$$\theta_r = \frac{0.5\left[\begin{array}{c} 5\theta_5 + (\theta_5 + \theta_{10}) * (10 - 5) + (\theta_{10} + \theta_{20}) * (20 - 10) \\ +(\theta_{20} + \theta_{40}) * (40 - 20) + (\theta_{40} + \theta_{60}) * (60 - 40) \\ +(\theta_{60} + \theta_{80}) * (80 - 60) + (\theta_{80} + \theta_{120}) * (120 - 80) \\ +(\theta_{120} + \theta_{160}) * (160 - 120) \end{array}\right]}{160}$$

We added the above information in the text (please see lines 308-312 in the revised manuscript). Besides, we revised the Eq.13 in the text as well.

L291-294: Is this the commonly used method to calculate NDVI? If so, you do not need to mention these details. And I have no idea why you describe the NDVI_Terra and NDVI_Aqua. Can you clarify?

Answer: Yes, this method is commonly used to calculate NDVI. As we found that there were slight differences ( $\left|NDVI_{Terra} - NDVI_{Aqua}\right| = 0.01 \pm 0.0075$ ) between the calculated daily $NDVI_{Terra}$ and $NDVI_{Aqua}$ , we calculated NDVI by averaging $NDVI_{Terra}$ and $NDVI_{Aqua}$ in our study in order to eliminate the impacts caused by such differences.

We added the above information in the section 3.2.2 (vegetation parameter), please see lines 272-277 in the revised manuscript. In addition, we followed your suggestion to simplify the descriptions of the method to calculate NDVI by firstly deleting the superfluous sentences to describe $NDVI_{Terra}$ and $NDVI_{Aqua}$ (e.g., L449-451, L453-461, L472-473 and L474-475 in the marked-up manuscript). Then we rephrased the sentences to state the method to calculate NDVI by averaging $NDVI_{Terra}$ and $NDVI_{Aqua}$ (please see lines 267-278 in the revised manuscript).

L398-399: Since NDVI is a normalized factor and you derived the NDVI_w based NDVI, I do not think it is meaningful to quantify the impact of NDVI on evaporation. This relationship might be changed in different cases and even with different time series datasets. You can describe this relation, but it is probably not suitable as a highlight and mention it in Abstract.

Answer: We agree with you that the relationship between NDVI and ET differs in different cases or with different time series. We also discussed this point in the manuscript (section 5.1). The main purpose for describing the relationship is to compare our result with other studies in different cases to show how strong NDVI affects ET. By our survey, the relationship between NDVI and ET were reported mostly in forests and grassland. Thus, it is meaningful to fill the gap to quantify the effect of NDVI on ET in shrubland.

In summary, we described this relationship in the main body for comparison, and following your suggestion, we deleted this description in the abstract.

L412: do you mean "compared Period I with. . ."?
Answer: Yes, we revised the sentence from "compared to period I with natural land use/cover condition, …" to "compared with Period I, …", please see line 426 in the revised manuscript.

L431-434: The first-order control of evaporation is a long time debate. I agree with the conclusion, but this research might be not directly related to this conclusion. I suggest the authors weaken the tone, to use "probably" or "very likely" etc.
Answer: We followed your suggestion and revised the word by replacing "mainly" in this sentence with "likely", please see line 445 in the revised manuscript.

L545: "tolerant to" is probably followed by some "vices", not survive. Please rephrase.
Answer: We revised the sentence from "…because shrubs are more tolerant to survive in water-starved ecosystems" to "… because shrubs are easier to survive in water-limited ecosystems", please see lines 545-546 in the revised manuscript.

L 550: more water than "what"?

Answer: The missing "what" in this sentence was the word "grass". However, we deleted this sentence "As potato consumes much more water than grass". Because we have emphasized the fact that potato consumes more water than grass in this paragraph, please see line 842-843 in the marked-up manuscript.

**Reply to Referee #2**

This study assessed the relationships between evapotranspiration (ET) and change of land by analyzing the eddy-covariance measurements of actual ET together with data of a number of its potentially influencing factors including normalized vegetation index, soil water content as well as climate variables to estimate potential ET. Data are collected from a case study with different periods reflecting changes of land-use conditions, which provides further evidence to support the statistical analyses. The manuscript is well written and the knowledge promoted is a clear contribution to the understanding of how ET processes can potentially change with land-use in semiarid regions. I think this study is suitable for publication after moderate revision, with improved clarity and better flow especially for the Introduction and Methodology - please see my major comments below.

Answer: Thank you for your positive comments. We revised our manuscript by carefully following your comments and suggestions.

**Major comments:**

**1.** The Introduction launched quite well with highlighting the importance of the assessing relationship between ET to vegetation conditions in arid/semiarid regions (L28-37), followed by a comprehensive literature review explaining relevant physical mechanisms (L38-61). However, the third paragraph (L62-79) seems to be a bit disjointed as the flow of ET/vegetation stops and shifts to the case study, whereas paragraph 4 (L80-98) returns to the ET/vegetation flow and paragraph 5 again introduces the study site. I think the easiest way to improve the flow is by swapping paragraph 3 and 4 (I found this can in fact fit better with your current connecting sentences between paragraphs i.e. L60-61, L96-98). So you would have: Paragraph 1: importance of the assessing relationship between ET to vegetation conditions in arid/semiarid regions Paragraph 2: physical mechanisms on how vegetation can influence ET (finish with L60-61 which then leads to the method of assessing these impacts) Paragraph 3: method to assess the vegetation impact on ET (finish with L96-98 which then leads to the case study in a sparse shrubland) Paragraph 4: introducing the case study and how it can contribute to the above-mentioned knowledge gap. I'd also recommend combining this use some discussion from the current paragraph 5 (L99-101) to help justifying the choice of the case study. Paragraph 5: I'd recommend to leave this paragraph purely as a summary of the study (as current L102-104), and maybe elaborate a little bit with highlighting the significance of the study. I think the above structure can allow the storyline about ET/vegetation relationship to complete before introducing the study site, which provides a smoother transition and also better justification on the use of Mu US sandland as the case study.

Answer: Thank you for your constructive suggestions. According to your comments, we swapped paragraph 3 and 4 to make these paragraphs jointed, please see lines 58-92 in the revised manuscript.

In addition, we also combined the following sentences to paragraph 4 to help justify the choice of the study site.

"Coincidentally, two processes of land use/cover changes (land degradation and vegetation rehabilitation) have occurred at the edge of the Mu Us Sandy Land, providing us a unique opportunity to study the effects of land use/cover change on ET."

Furthermore, in paragraph 5, we highlighted the significance of our study by adding the sentence "Our results were expected to provide a scientific reference for the sustainable management of regional land and water resources in the context of intensive agricultural reclamation", please see lines 95-97 in the revised manuscript.

**2.** I appreciate the comprehensiveness of Section 2 which covers the details of data collections methods and models used to analyze different data variables. However, I found that Section 2.3.2 become a bit confusing with introducing models related to a number of variables. As this section describes the methods employed for the core analyses of the study, I think the clarity can be further improved by using further subsections for individual variables. In addition, I think the methods used for data analyses should be introduced as well. Currently the statistical methods used for data analyses are mainly described in the Results section (e.g. L325-327, L380-384, L386-L389). I think it can be clearer to summarize them in the Section 2.3.2 instead (probably as an overview in the start of this section). In this way you can better justify why these analyses are conducted and how they help to answer the research questions, while purely focusing on the results and interpretation in the Results section. And then the readers can get an overall understanding on the data analyses to be conducted and knowing what to expect in the Results section. So I'd suggest the following structure for Section 2.3.2: Sub-section 1: overview – introducing the variables which are needed for analyzing the impact of ET and vegetation conditions (these will be detailed in the following sub-sections), and what analyses will be conducted with these variables (e.g. as those introduced in L325-327, L380-384 and L386-L389 etc.) Sub-section 2: estimating potential ET Sub-section 3: estimating soil water content Sub-section 4: estimating NDVI ...

Answer: Together based on your comment 2 and 3, we separated the pervious section 2 into two sections (section 2 and section 3). In the new section 2, we mainly introduced the case study information (including site information, the measurements in our study site) and data (we thought the data here included raw data and processed data). Thus, we thought it might be better to move the previous section 2.3.1 (flux data processing) into this new section 2 as subsection 2.3. We re-arranged and revised section 2 based on the following structure:

Case study and data

…2.1 Site description

….2.2 Field measurements

……2.2.1 Eddy covariance system

……2.2.2 Other measurements

…2.3 Flux data processing (lines 178-212)

While in section 3, we mainly introduced the methods to calculate the footprint and the variables that have controls on ET. Following your suggestion, we added a subsection 3.3 (statistical analysis), including the statistical methods that described in the Results section previously (as you referred in this comment 2, e.g., L380-384, L386-L389 in previous manuscript). In addition, we also added the information about the reason why we chose linear function to simulate the correlations between ET and its three controlling factors (please see our reply to the minor comment 12).

For the sentence you mentioned in this comment 2 (e.g., L325-327 in the previous manuscript), which described the purpose to calculate energy balance closure, we prefer to leave them in the Result Section (lines 343-346). Because energy balance closure is a common concept and there is no need to describe it in the method part. In addition, if we move the sentence to the method part, the continuity of section 4.1 will be broken.

Therefore, we re-arranged and revised the section 3 according to the following structure:
Methodology
…3.1 Footprint model
…3.2 Method of analyzing controlling factors of ET
……3.2.1 Potential evapotranspiration
……3.2.2 Vegetation parameters
……3.2.3 Soil water stress
…3.3 Statistical analysis (new subsection, lines 320-329).

**3.** I think the Section 2 (Material and Methods) is a bit too long trying to cover different aspects including case study, measurements of raw data, data processing and analyzing. In my opinion a better way to organize these is to break Section 2 into two sections, for example as: Section 2. Case study and data (note:I'd use 'data' to refer to the raw measurements here rather than in the next section, where you introduce data-processing and analyzing.) 2.1 site description 2.2 measurements ... Section 3. Methodology 3.1 flux data processing 3.2 footprint model 3.3 method of analyzing controlling factors of ET (and if you agree with my last comment, the sub-sections can go below:) 3.3.1 ... 3.3.2 ... ...
Answer: Please see our reply to the comment 2.

**Minor comments:**
**1.** L30: 'ET' - please define acronym when it first appears in the text, and please also check if all other acronyms are properly defined.
Answer: We added the definition of ET in the text, please see line 26 in the revised manuscript.

**2.** L101: '4' - please spell out numbers less than 10 i.e. as 'four-year'.
Answer: We revised it.

**3.** L111: please delete the repeated 'temperate'. Also, is there a better way to introduce the climate zone, as currently it seems like a 'noun train' ('temperate semiarid continental monsoon climate'). You can find some examples on improving 'noun train' from http://www.webwritingthatworks.com/DGuideCOG5b.htm.
Answer: We have studied the guidelines from the link you provided. Together based on the guidelines and other scholars' studies (Yang et al., 2015; Wu and Ci, 2002). We thought the sentence might be better by changing it from "the study site is in a temperate semiarid continental temperate monsoon climate" to "This site is a semiarid area with temperate continental monsoon climate", please see line 104 in the revised manuscript.

References:
Yang, Y., Bu, C., Mu, X., Zhang, K.: Effects of differing coverage of moss‐dominated soil crusts on hydrological processes and implications for disturbance in the Mu Us Sandland, China. Hydrological Processes, 29(14), 3112-3123, 2015.
Wu, B., Ci, L. J..: Landscape change and desertification development in the Mu Us Sandland, Northern China. Journal of Arid Environments, 50(3), 429-444, 2002.

**4.** L194-195: Would there be any impact on the results from this data removal, and would this be a limitation of the study? This should be briefly discussed (Maybe in the Discussion or Conclusion section?).
Answer: We thought there might be little impact of this data removal on our results due to the following reasons.

Firstly, in our study, the missing and rejected $\lambda E_T$ values almost occurred during nighttime (89.1% in Period I, 91.3% in Period II, 92.6% in Period III and 88.7% in Period IV), which were mainly caused by insufficient electric power supply in low air temperature environment and the low turbulence during the nighttime.

Secondly, in the nighttime, the change in $\lambda E_T$ is small, and ET values are close to zero. Therefore, after removal of the nighttime data, the errors of the gap-filled nighttime values based on the neighboring good data are small. Besides, $\lambda E_T$ values of nighttime accounted a very small proportion to the daily ET.

Thirdly, the ratios of the missing and rejected data points are not so high. For example, Falge et al. (2001) have reported that during quality control procedure of 28 flux sites, there was an average of 31% missing or rejected values of $\lambda E_T$ values. Wever et al. (2002) reported that there was 15% missing or rejected values of $\lambda E_T$ values during the quality control procedure. Mauder et al. (2006) have reported that there was an average of 20% missing or rejected values of $\lambda E_T$ values by 20 flux sites. Therefore, the ratio of rejected and missing half-hourly data in each period was reasonable and the dataset of $\lambda E_T$ after quality control procedure is reliable.

We added the above reasons in the text, please see lines 188-197 in the revised manuscript.

References:
Falge, E., Baldocchi, D., Olson, R., Anthoni, P., Aubinet, M., Bernhofer, C., Burba, G., Ceulemans, R., Clement, R., Dolman, H., Granier, A., Gross, P., Grunwald, T., Hollinger, D., Jensen, N. O., Katul, G., Keronen, P., Kowalski, A., Lai, C. T., Law, B. E., Meyers, T., Moncrieff, H., Moors, E., Munger, J. W., Pilegaard, K., Rannik, U., Rebmann, C., Suyker, A., Tenhunen, J., Tu, K., Verma, S., Vesala, T., Wilson, K. and Wofsy, S.: Gap filling strategies for long term energy flux data sets. Agricultural and Forest Meteorology, 107, 71-77, 2001.
Mauder, M., Liebethal, C., Göckede, M., Leps, J. P., Beyrich, F., Foken, T.. Processing and quality control of flux data during LITFASS-2003.Boundary-Layer Meteorology, 121(1), 67-88, 2006.

Wever, L. A., Flanagan, L. B., & Carlson, P. J. (2002). Seasonal and interannual variation in evapotranspiration, energy balance and surface conductance in a northern temperate grassland. Agricultural and Forest Meteorology, 112(1), 31-49.

**5.** L208-211: It would be clearer if these lines can be presented as individual formulae (i.e. in the format of L219). Also, according to L205, the 'n' in 'Rn' should be subscripted - please also check that the use of other symbols is consistent throughout the text.

Answer: After considering your comment, we thought it might be better to change these lines from several formulae to the following form: " $f = a * (R_n - G)^2 + b * (R_n - G) + c$ (Period I: a = 0.0014, b = 0.075, c = 10.69, R = 0.77; Period II: a = 0.0012, b = 0.056, c = 17.69, R = 0.67; Period III: a = 0.0014, b = 0.16, c = 13.24, R = 0.75; Period IV: a = 0.0015, b = -0.083, c = 25.87, R = 0.69)" , please see lines 206-209 in the revised manuscript.

In addition, we revised the $R_n$ and checked the use of other symbols throughout the text.

**6.** L246: 'psychrometric constant' - what is the value of the constant?

Answer: We added the equation of psychrometric constant (Eq.6 in the revised manuscript), please see lines 252-255 for detailed information.

**7.** L248: 'U2' - where is it in Equation (5)?

Answer: $U_2$ is used to calculate the aerodynamic resistance ($r_a$, Eq.7 in the manuscript). We moved the equation of calculating $U_2$ from Eq.6 to Eq.8 for better understand, please see lines 259-262 for detailed information.

**8.** L337: 'Ds' - not defined as in Minor comment #1. Also, how are the data of Ds obtained? I couldn't seem to find it in Section 2.2.2 (other measurements).

Answer: Thank you for your kind remind. We added the information of measurement that obtained $D_s$ in the section of "other measurements" with the following sentence. "Sunshine duration ($D_S$) is measured by a sunshine recorder (CSD3; KIPP&ZONEN, Delft, the Netherlands)."

**9.** L337: 'normal' - I think 'average monthly' would be a better description here.

Answer: We revised it.

**10.** L347: Figure 4 has not been introduced in the text yet, should it be mentioned somewhere between L336-337?

Answer: Yes, we added the sentences of "Four-year and long-term (1954-2014) average monthly values of $D_s$, $T_a$, $R_H$, and P are shown in Fig.4." in the section 4.2, please see lines 356-358 in the revised manuscript.

**11.** L380: 'relationships' - 'correlations' would be a more accurate description.

Answer: Thank you for your suggestion. We revised it, please see line 403 in the revised manuscript.

**12.** L389-390: the r2 only investigates linear relationships - are you expecting any non-linear relationships which are not covered here and would this be a limitation? This can be briefly discussed.

Answer: We already used several common functions (e.g., exponential function, linear function, logarithmic function and quadratic function) to fit the correlations between ET and its controlling factors ($E_{TP}$, NDVI and $f_s$). The values of determination coefficient ($R^2$) are listed in the following Tab. 1.

According to the results that showed in the following Tab.1, $R^2$ of the linear function is generally the highest. Therefore, we chose the linear function to fit the correlations between ET and its three influencing factors in our study.

We added the above information in the section 3.3 (Statistical analysis), please see lines 321-326 in the revised manuscript.

Table 1. The determination coefficient ($R^2$) of the correlations between ET and the three controlling factors.

|  | ET and $E_{TP}$ | ET and NDVI | ET and $f_s$ |
|---|---|---|---|
| Exponential function | 0.46 | 0.52 | 0.27 |
| Linear function | 0.46 | 0.52 | 0.28 |
| Logarithmic function | 0.43 | 0.51 | 0.19 |
| Quadratic function | 0.45 | 0.51 | 0.28 |

**13.** L464: The term 'BSC' has already been defined in L68.

Answer: We deleted the definition here.

**14.** L565: It should be worth highlighting some significance and contributions of this study towards the end of conclusion.

Answer: We highlighted the significance of our study at the end of conclusion with the following sentences:

"Furthermore, our results suggest that when we simulate the impact of land use/cover change on hydrological processes, vegetation factor might not be the unique factor to parameterize, instead, the integrated effects of land surface and vegetation conditions should be considered. Our study also provides a scientific reference to the regional sustainable management of water resources in the context of intensive agricultural reclamation.", please see lines 566-571 in the revised manuscript.

**15.** Fig. 6: I don't think the use of different shapes is necessary given that you are using multiple panels?

Answer: We revised Fig.6.

**16.** L884 (title of Fig. 6): 'r: Pearson's correlation significance' should r be 'Pearson's correlation coefficient' instead?

Answer: Yes, we revised it in the title of Fig.6, please see line 944 in the revised manuscript.

Monitoring the variations of evapotranspiration due to  land use/cover change in a semiarid shrubland

Tingting Gong, Huimin Lei, Dawen Yang, Yang Jiao, Hanbo Yang

State Key Laboratory of Hydroscience and Engineering, Department of Hydraulic

Engineering, Tsinghua University, Beijing, 100084, China

**Correspondence to**: Huimin Lei (leihm@tsinghua.edu.cn)

**Abstract**

Evapotranspiration ($E_T$) is an important process in the hydrological cycle, and vegetation change is a primary factor that affects $E_T$. In this study, we analyze the annual and inter-annual characteristics of $E_T$ using continuous observation data from eddy-covariance (EC) measurement over four  years (

1 July 2011 to 30 June 2015)  in a  semiarid shrubland

Mu Us Sand Land  China. The  normalized difference vegetation index (NDVI) was demonstrated  as the predominant factor that influences the seasonal variation in $E_T$.

Normalized $E_T$

Additionally,

Both on during the  land degradation  and vegetation rehabilitation process,  $E_T$ and normalized $E_T$ both increased due to the integrated effect of the changes in vegetation type, topography, and soil surface characteristics.

This study could improve our understanding of the effects of land use/cover changes on $E_T$ in the fragile ecosystems of semiarid regions and provide a scientific reference for the sustainable management of regional land and water resourcesimprove our understanding

s of semiarid regions.

**Key words**: evapotranspiration; normalized difference vegetation index

; land use/cover change; eddy covariance; semiarid region

**1 Introduction**

Arid and semiarid biomes cover approximately 40% of the Earth's terrestrial surface (Fernández, 2002). Previous studies have shown that more than 50% of precipitation ($P$) is consumed by evapotranspiration ($E_T$) (Yang et al., 2007; Liu et al., 2002). Moreover, a slight change in $E_T$ could have significant influences on water cycle and  the ratio of $E_T/P$ could increase to even 90% or more in these regions (Mo et al., 2004; Glenn et al., 2007).  In terms of physical processes, $E_T$ is affected by climatological factors (e.g., direct solar radiation, air temperature, vapor pressure deficit and wind speed) but also vegetation conditions. For example, direct solar 
[revised manuscript text omitted]
 ~~were contained in this software. The quality control was performed on the half-hourly output files, and calculated $\lambda E_T$ was flagged as 0 (excellent quality), 1 (good quality) and 2 (bad quality, removed and need to be gap-filled), respectively. The basic principle of the technique is that flux is calculated ?????. The software provides almost all the essential correction procedures including (1) detection and elimination of spikes; (2) tilt correction; (3) sensor separation correction; (4) density fluctuation correction (Webb et al., 1980). The calculated half-hourly flux datasets were further filtered for the remaining spikes, instrument malfunctions, and poor quality, according to the following criteria (Papale et al., 2006): (1) incomplete half-hourly measurement, mainly caused by power failure or instrument malfunction; (2) rainy events; and (3) outliers caused by occasional spikes for unknown reasons.ratiosrejectedare~~were 17.3% in Period I, 20.2% in Period II, 16.5% in

Period III, and 18.6% in Period IV., and Aalmost all missing and the rejectedremoved $\lambda E_T$ values occurred during the nighttime (89.1% in Period I, 91.3% in Period II, 92.6% in Period III, and 88.7% in Period IV). During the nighttime, the change in $\lambda E_T$ was small, and$E_T$ values were close to zero., and besides, $\lambda$ Therefore, after removal of the nighttime$\lambda E_T$ values, the errors of the gap-filled nighttime values based on the neighboring good data were small. Besides, nighttime $\lambda E_T$ values accounted for only a small proportion of the daily $E_T$. Furthermore, Tthe ratiopercentages of rejected and missing data in our study are close to other scholars' results. The reported percentage was summarized in a range of 15%~31% (For example, Falge et al. (, 2001; Wever et al., 2002; Mauder et al., 2006) have reported that during quality control procedure, there was an average of 31% missing or rejected values of $\lambda E_T$ values by 28 flux sites. Wever et al. (2002) reported that there was 15% missing or rejected values of $\lambda E_T$ values during the quality control procedure. Mauder et al. (2006) have reported that there was an average of 20% missing or rejected values of $\lambda E_T$ values by 20 flux sites. In addition, $\lambda E_T$ values during nighttime changed steady and close to zero, coupling with the fact that they accounted a very small proportion throughout whole day. Therefore, the dataset of $\lambda E_T$ after quality control procedure was isconsidered reliable to use.

[revised manuscript text omitted]

source area were chosen to calculate NDVI.

$$NDVI = \frac{NIR-VIS}{NIR+VIS} \tag{9}$$

$$NDVI_{Terra\ or\ Aqua} = \frac{NIR-VIS}{NIR+VIS}$$

where

NIR is the spectral response in the near-infrared band (857 nm ) and VIS is the visible red radiation band (645 nm).

convert the daily Surface Reflectance data to the Universal Transverse Mercator

In this study, NDVI was calculated by using MODIS/Terra data (MOD09GQ)

( $NDVI_{Terra}$ ) and MODIS/Aqua data (MYD09GQ) ( $NDVI_{Aqua}$ )

(http://reverb.echo.nasa.gov), respectively. As we found that there were slight differences ($\left|NDVI_{Terra} - NDVI_{Aqua}\right| = 0.01 \pm 0.0075$) between $NDVI_{Terra}$ and

$NDVI_{Aqua}$, we calculated NDVI by averaging $NDVI_{Terra}$ and $NDVI_{Aqua}$ in order to eliminate the impacts of such differences.

T, he calculated NDVI values  were then filtered to remove anomalous  hikes and drops (Lunetta et al., 2006).

, and the  smoothing spline method was used to produce a smoother profile.

Terra Aqua

Theoretically, land use/cover changes can be evaluated by comparing the land use/cover maps in two different periods. However,  transient land use/cover maps were unavailable at our site. Therefore, we separated the study area within the footprint  into two zones:  the undisturbed zone without any land use/cover changes was deemed as zone A and  the disturbed zone with land use/cover changes was deemed as zone B. In zone A, vegetation  changes included only vegetation phenological changes; however, in zone B, there were not only vegetation phenological changes but also land use/cover changes. Based on the assumption that the phenological changes caused by climate in the two zones were the same, we defined an indicator ($D_{lu}$) as a measure of land use/cover changes:

$$D_{lu} = M_A - M_B \tag{10}$$

where $M_A$ and $M_B$ are the monthly vegetation coverages of zone A and zone B, respectively. The monthly vegetation coverage was calculated by monthly NDVI values (Gutman and Ignatov, 1998):

$M = (\text{NDVI} - \text{NDVI}_{min})/(\text{NDVI}_{max} - \text{NDVI}_{min})$ (11)

where $\text{NDVI}_{max}$ is the maximum value (0.8 in this study) and $\text{NDVI}_{min}$ is the minimum value (0.05 in this study) (Gutman and Ignatov, 1998). The calculated monthly M values (27.6% and 24.2%) were consistent with the measured vegetation coverages in August 2011 (28.2%) and September 2011 (27.9%) at our study site.

3.2.3 Soil water stress

The effects of the soil water content on $E_T$ can be described in three stages (Idso et al., 1974), stage 1: the soil water is enough to satisfy the potential evaporation rate ($f_s$=1); stage 2: the soil is drying and water availability limits $E_T$ (0<$f_s$<1); and stage 3:

the soil is dry and evaporation can be considered negligible ($f_s$=0). We used daily soil water content in the root depth ($\theta_r$) to estimate $f_s$ by the following expression (Hu et al., 2006):

$$f_s = \begin{cases} = 1 & \theta_r > \theta_k \\ = 0 & \theta_r < \theta_w \\ = \frac{\theta_r - \theta_w}{\theta_k - \theta_w} & \theta_w \leq \theta_r \leq \theta_k \end{cases}$$ (12)

where $\theta_w$ is the wilting value and $\theta_k$ is the stable field capacity which is considered to be equivalent to 60% of the field capacity (Lei et al., 1988; Wang et al., 2008). $\theta_r$

was calculated by measured soil water contents at different depths ($\theta_i$; $i$ =5, 10, 20,

40, 60, 80, 120 and 160 cm). From land surface to the depth of 5 cm, the soil water profile was regardedassumed as a triangletriangular, while inat other depths, the soil water profiles were treatedassumed as trapezoidstrapezoidal. Therefore, the root zone soil moisture of root zone was calculated equation was as,:

$$\theta_r = \frac{0.5\begin{bmatrix} 5\theta_5+(\theta_5+\theta_{10})*(10-5)+(\theta_{10}+\theta_{20})*(20-10) \\ +(\theta_{20}+\theta_{40})*(40-20)+(\theta_{40}+\theta_{60})*(60-40) \\ +(\theta_{60}+\theta_{80})*(80-60)+(\theta_{80}+\theta_{120})*(120-80) \\ +(\theta_{120}+\theta_{160})*(160-120) \end{bmatrix}}{160} \tag{123}$$

where, the sSite-averaged soil water content of each depth ($\theta_i$; ($i =$5, 10, 20, 40, 60,

80, 120 ,and 160 cm) was calculated by taking a weighted average of the measured values in the canopy and bare surface patches,

$\theta_i = M \times \theta_{i,c} + (1 - M) \times \theta_{i,b}$

(134)

where $\theta_{i,c}$ and $\theta_{i,b}$ refer to the measured soil water contents of canopy patch and bare soil patch at the depth of $i$ cm, respectively; $M$ M is the monthly vegetation coverage of undisturbed zone, and it was calculated by monthly Normalized Difference

Vegetation Index (NDVI) values (Gutman and Ignatov, 1998),

$M = (NDVI - NDVI_{min})/(NDVI_{max} - NDVI_{min})$ (14)

where $NDVI_{max}$ is the maximum value (0.8 in this study); $NDVI_{min}$ is the minimum value (0.05 in this study) (Gutman and Ignatov, 1998). The calculated monthly $M$ (27.6%

and 24.2%) was consistent with the measured vegetation coverage in August 2011

(28.2%) and September 2011 (27.9%) at our study site..

3.3 Statistical analysis

In this study, we chose

daily data in period I

to conduct the linear regression between available energy ($R_n$ ) and the sum of surface fluxes ($\lambda E_T + H$), thus is used to validate EC measurements and examine the quality of flux data.  daily data in Period I

to analyze the correlations between $E_T$ and the three controlling factors ($E_{TP}$,

NDVI , and $f_s$). We used several common functions (e.g., an exponential function, a linear function, a logarithmic function and a quadratic function) to fit these correlations. We found that the determination coefficient ($R^2$) of the linear function was generally the highest.

Therefore, in this study, we chose the linear function to fit the correlations between $E_T$ and the three controlling factors. Additionally, significant t-test was performed to evaluate the degrees of these correlations

. Moreover, Ddata on rainy days was removed because  $E_T$ values were gap-filled rather than measured

4 Results

4.1 Footprint and energy balance closure

  Based on the footprint model, we got the half-hourly scatter data (Eq. 2), and according to the wind rose diagram (Fig. 3a), the prevailing wind directions  at this site were northwest and southeast. Therefore,  we chose an ellipse to enclose the scatters and simulated the footprint (Fig. 3b).

There  was 93% half- hourly flux data within the ellipse under unstable conditions.

  Additionally,  we measured the boundary of zone B in October 2013 when the land use/cover condition in zone B had stopped  change (Fig. 3b). There were 11

pixels (250 m $\times$ 250 m) in zone A and 19 pixels (250 m $\times$ 250 m) in zone B, and thus, in the following part of calculating the weight-averaged NDVI

( $NDVI_w$ ) within the footprint fetch, we chose the weighted coefficient as $\beta =$

$11/(11 + 19)$.

[Figure 3 is to be inserted here]

  EC system performance was assessed by the energy balance closure which was calculated by conducting the linear regression between available energy ($R_n - G$) and the sum of surface fluxes ($\lambda E_T + H$), which is also used to examine the quality of flux data (Wilson et al., 2002).

The linear regression yielded a slope of 0.87, an intercept of -1.42 W m$^{-2}$, and an $R^2$ of 0.82. These indicators  suggested that the measurements at our experimental site provided reliable flux data, and that the EC measurements underestimated the sum of the surface fluxes to the extent of 13%. A lot ofMany researchers have investigated the energy imbalance (Barr et al., 2006; Wilson et al., 2002; Franssen et al., 2010), and there is a consensus that it is difficult to examine the exact reasons leading to thefor the imbalance.

34.2 Characteristics of environmental variables

A brief summary of the key environmental variables will beisis presented in this section. Four-year averaged monthly sunshine duration ($D_s$), $T_a$, $R_H$, $P$ and long-term (1954-2014) averaged monthly values of $D_s$, $T_a$, $R_H$, and $P$ wereare showed in Fig. 4.

Monthly $D_s$$D_s$ was much higher than the long-term normal average monthly values, of

1954-2014 except in July and September. The highest value of monthly $D_s$$D_s$ was observed in May (299.5 h) and the lowest was observed in February (206.6 h). The

Seasonal seasonal characteristics of $T_a$Ta showed a highly similar trend with that of the long-term average monthly values of 1954-2014normal, and the differences were less than 1 ℃, ℃ except in July, January and March. The highest value of monthly

$T_a$Ta was observed in July (22.1 ℃℃) and the lowest was observed in December (-

8.1℃℃). The values of $R_H$RH showed were almost lower than the long-term average monthly values of 1954-2014normal, especially in March and April. The highest $R_H$$R_H$

was observed in September (65.4%) and the lowest was observed in March (35.1%).

The seasonal distributions of $P$ were consistent with the long-term average monthly values of 1954-2014normal, and 89.7% of $P$ happened occurred in the growing season.

The value of $P$ was highest in July was the highest (120.5 mm) and in January was the lowest in January (0.3 mm).

[Figure 4 is to be inserted here]

The inter-annual characteristics of daily $T_a$, $D_s$, $R_H$,  $\theta_r$, groundwater level (GWL), and total $P$ in the growing season of each period  are listed in Tab.

1.

[Table 1 is to be inserted here]

The values of $T_a$ , $R_H$ , $P$, and $\theta_r$ in the growing season of Period IV were the lowest compared  to those in other three periods. Period I-~III  were all wet year, while Period IV was  a dry year. The values of $\theta_r$ in Period I-~III were similar, however, $\theta_r$ decreased by 0.0113 m³ m⁻³ in Period IV. The mean GWL in Period III was the shallowest.

4.3 Seasonal variations in $E_T$ due to climate variability

The  seasonal curve of $E_T$ in each year had a single peak value (Fig. 5a), with  higher $E_T$ appearing mostly in the growing season while  lower appeared in the non-growing season. The daily $E_T$  range from 0.0 mm day⁻¹ to 6.8 mm day⁻¹ during the four periods, the highest $E_T$  was observed on 22th June 2013

, which was the day after a continuoal rainfall event  started from 19th June 2013 to 21th June 2013 (90.3 mm),

. The lowest $E_T$ appeared on 28th November

2012, which was in the frozen period (late November to early March  at our study site).  On rainy days, $E_{TP}$ (Fig. 5b) was  lower due to lower net radiation and air temperature. $E_{TP}$  range  from 0.2 mm day⁻¹  in December

2011 to 17.9 mm day$^{-1}$  in September 2013.

The  seasonal NDVI curve  for natural land use/cover condition (in zone A during Period I IV and in zone B during Period I) represented the process of natural vegetation phenology and it had  a single peak value in each year (Fig. 5c).

In early May, the seasonal NDVI curve began to increase asand the native vegetation  entered the growing season, and a maximum value (0.27±0.01) was reached in July or August. In the winter, the daily

NDVI remained relatively constant (0.13±0.01). $f_s$ (Fig. 5d)

increased rapidly in response to rainfall events of more than 5 mm a day and decreased rapidly one or two days after rainfall events.  From late

November to early March, there was a frozen period at this site, and soil water content was below the wilting point. The groundwater level  changed obviously in the monsoon season (July to September) and mildly in the winter (December to

February).

The linear correlations between $E_T$ and the three controlling factors  all passed the  *t*-test at a 95% confidence level. The value of the correlation between $E_T$ and $NDVI_w$ (  ($NDVI_w =$

$NDVI_A \times \beta + NDVI_B \times (1 - \beta)$)) was the largest, indicating that NDVI was highly correlated with the daily variations  in $E_T$. To better quantify the effects of the phenological process on $E_T$, the correlation between daily  and $NDVI_w$ in Period I  was analyzed (Fig. 7a).

[Figure 7 is to be inserted here]

A  positive linear regression was found between $f_v$

and $NDVI_w$ (Fig. 7a). The slope of the linear regression was used to evaluate the controlling degree between $f_v$  and vegetation phenological process, which.

indicating that when $NDVI_w$  increased one unit, it would contribute $f_v$  to increase about 1.86 units.

4.4 Inter-annual variations in $E_T$ due to land use/cover changes

During the four periods, in zone A, the NDVI values of each period were similar because the land use/cover condition was not changed. While in zone

B, the peak values of NDVI first  declined from 0.28 to 0.15 (Period I to Period III)

due to the  land use/cover  condition changed from mixed vegetation to bare soil. The peak NDVI value then increased to 0.22

(Period IV) due to  grass recovery (Fig 5c). An interesting phenomenon was found accompanied by the changing process of land use/cover condition: $E_T$ in the growing season  gradually  increasing from Period

I to III (Tab.2) while it  increased  greatly in Period IV even with less precipitation, because a mass of soil water and ground water was consumed to satisfy the $E_T$ demand (Fig. 5e).

[Table 2 is inserted to be here]

Compared  with Period I , $D_{lu}$ values of

Period II and Period III gradually increased,  while $D_{lu}$ of Period IV decreased.

Taking August in each period as an example, in  Period I, $D_{lu}$ was 0.2%, while in  Period II IV, $D_{lu}$ were 2.9%, 12.6%, and

8.6%, respectively. In order to eliminate the influence of vegetation phenological change on $E_T$, we chose the growing season of each period to analyze the correlation between $f_v$ and $D_{lu}$.

Quantitative results of the correlation between $D_{lu}$ and

  are shown in Fig. 7b.

From Period I to Period III,  with the changed land surface characteristics (the natural vegetation in  zone B was cleared ~~(Period I~III),,were~~ were disappeared)

~~(Period I~III), normalized $E_T$ (i.e., $f_v$)~~ increased and the increment was more evident in Period III (from 78.5 to 88.1). When the land use/cover condition  in zone B

gradually changed from bare soil to sparse grassland due to the self-restoring capacity of nature,  ($f_v$ )increased  significantly (from 88.1 to 111.3).

Discussion

5.1 Implications of the  effects of phenological change on $E_T$

The correlations between $E_T$ and its controlling factors infer that at our experimental site, NDVI was the predominant factor that influence the seasonal variation  in $E_T$. The  positive linear relationship between $f_v$

and NDVI  suggested that transpiration  was probably  controlled by the stomatal conductance and the number of stomata, which  are proportional to leaf area (Pearcy et al., 1989; Turrell, 1947), rather than the atmospheric water demand represented by $E_{TP}$.

Various studies have  assessed the  correlation between vegetation phenological change and $E_T$, and these results generally showed consistent and positive linear relationships (Nouri et al., 2014; Rossato et al., 2005; Duchemin et al., 2006; Glenn et al., 2008). However,  for different vegetation species, phenological change have effects on $E_T$  to different degrees. Relative strong regressions between NDVI and $E_T$ have been reported at forested sites (Loukas et al.

2005; Nouri et al., 2014; Chong et al., 2007) and grass-covered sites (Kondoh and

Higuchi, 2001; Nouri et al., 2014), with

 determination coefficients  higher than 0.7,  reveal the strong control of phenological change on $E_T$.

Thus, we speculate that, for high  vegetated ecosystems, phenological change  may have a strong and significant control on $E_T$. However, in low  vegetated ecosystems such as sparse shrubland in this study, the relationship between $E_T$ and seasonal vegetation phenological change is thus positive but relative weak.

5.2 Possible reasons for the effects of land use/cover changes

During Period IIV, the land use/cover condition  at our experimental site underwent two processes:  land degradation process (Period IIIII)

and vegetation rehabilitation process (Period IV). Interesting phenomena  were found  during these two processes: (1) $E_T$ and normalized $E_T$ values  both increased and (2)  normalized $E_T$ increased much faster during the vegetation rehabilitation process than that during the land degradation process.

The impact of phenological change on $E_T$ demonstrated that $E_T$ would decrease along with the leaf browning. Thus, we expect that $E_T$  would also decrease if leaves were cleared by human activities. However, during Period IIII, not only leaves were cleared, but also other land surface properties (all branches were cut off, sand dunes (fixed and semi-fixed) were bulldozed, and the dry sand layers and the biological soil crusts (BSCs) were destroyed) were changed, making resulting in the complex land use/cover condition complexs. All these changed land surface properties might contribute to the increase of $E_T$. The exists of dry sand layers and BSCs were demonstrated to effectively restrained the soil evaporation rates (Wang et al., 2006; Lv et al., 2006; Liu et al., 2006; Chen and Dong, 2001; Yang et al., 2015; Fu et al., 2010;

Liu, 2012). However, the bulldozing of sand dunes at our experimental site made the elevation of the flat soil surface be lower than the average elevation of the undisturbed soil surface (aboutapproximately 1.5 m lower, Figure 2(d). 2d), which resulted thatmaking the groundwater depth was much shallower than beforethe pre-disturbance depth. Thus, it is was hard for the formation of dry sand layers with shallower groundwater depthlevel. In this situation with the destroyed BSCs and the disappeared dry sand layers, the sufficient groundwater supply (Li and Li, 2000) accelerated the loss of water that stored in shallow soil through evaporation. The enhancement effect of soil evaporation offsetoffset the inhibition inhibiting effect of transpiration by due to leaves clearing, which made $E_T$ increase.

A secondary reason for the enhancement increase of soil evaporation was that more solar radiation was absorbed by soil layerthe soil layer absorbed more solar radiation during the land degradation process. In Period I, with natural vegetation, the radiation absorbed by the shadowed soil was the solar radiation transmitted into the canopy of shrubs and grass. However, with when the natural vegetation being was cut offcleared, the leaves and the branches were also removed, which made the shadowed soil exposed and enhanced the radiation absorbed by the soil, thereby increasing  soil evaporation (Martens et al., 2000; Panferov et al., 2001).

Moreover, the removal of leaves and branches and the disappearance of sand dunes both altered the land surface albedo could directly alter the solar radiation absorbed by the land surface (Dirmeyer and Shukla, 1994; Greene et al., 1999), subsequently leading to the change in $E_T$.

SSome inconsistent results regarding the  $E_T$ dynamics during during land degradation process were reported. A

portion of studies reported that $E_T$ decreased during the land degradation process with different reasons

. For example,

Souza and Oyama (2011) and

Snyman (2001)

demonstrated that $E_T$

decreased during the land degradation process due to  decreased transpiration in semiarid regions.

Lu et al. (2011) considered that

the low soil water content was thought to be main reason for the decrease of $E_T$ in the land degradation process.

Mao and Cherkauer (2009) also reported a decrease of  $E_T$

when land use/cover condition was  converted from forest to grass or cropland in the Great Lakes region. However, contrasting results ere also reported regarding the effects of land degradation on $E_T$  Hoshino et al. (2009)  found that there was no difference in $E_T$ during the land degradation  process associated with overgrazing in a semi-arid Mongolian grassland, and they  hypothesized that the reason for this lack of change might be the short grazing time (2 years).  Li et al. (2013) demonstrated that the warming air temperature was the main cause of the enhanced $E_T$ during the land degradation process in Qinghai-Tibet Plateau. Throughout the above researches 
[revised manuscript text omitted]

periods at the study site and climatological  normalmonthly average ( of 1954-2014 from climate ological normal in Yulin meteorological station).

Fig. 5. Seasonal and inter-annual characteristics of daily (a) evapotranspiration ($E_T$, mm); (b) potential evapotranspiration ($E_{TP}$, mm); (c) NDVI in zone A and zone B within the the footprint; (d) precipitation ($P$, mm); (ed) the soil water stress of the root zone ($f_s$) and (e) the groundwater level (GWL, m)  duringfrom 1st-1 July 2011 to 30th-30 June 2015.

Fig. 6. The correlations between daily evapotranspiration ($E_T$, mm) and its controlling factors: (a) daily potential evapotranspiration ($E_{TP}$, mm); (b) daily weight-averaged NDVI ($NDVI_w$) within the footprint (NDVI$_w$); (c) daily soil water stress of the root zone ($f_s$) in Period I by excluding the data in on rainy days (r: Pearson's correlation significancecoefficient; T: Tt-test significance).

Fig. 7. Quantitative analysis between of (a) the correlations between (a) vegetation phenological change ($NDVI_w$) and daily normalized $E_T$ ($f_v = E_T/(E_{TP} \times f_s)$) in Period I (excluded the data in on rainy days and frozen days); ) and (b) the indicator of land use/cover change ($D_{lu}$) and total normalized $E_T$ ($f_v = E_T/(E_{TP} \times f_s)$) of in the growing season in of each period.

Table 21. Daily air temperature ($T_a$ , ℃ ), relatively humidity ($R_H$ , %), total sunshine duration ( $D_S$ , h), soil water content of the root zone ($\theta_r$, m$^3$ m$^{-3}$), the groundwater level (GWL, m) and total precipitation ($P$, mm) in 1954-2014 and in the growing season of each period ( because there were some missing data in Period IV (from 12$^{th}$ September 2014 to 23$^{th}$ November 2014 and from 13$^{th}$ March 2015 to 22$^{th}$ April 2015), we  excluded data in these two time range of Period I III and 1954-2014)

| Variable | 1954-2014 | I | II | III | IV |
|---|---|---|---|---|---|
| $T_a$  (℃) | 19.8 | 19.6 | 20.4 | 19.9 | 19.3 |
| $R_H$  (%) | 57.7 | 57.3 | 54.9 | 53.4 | 52 |
| $D_S$   (h) | 213.3 | 220.7 | 215.8 | 218.2 | 220.7 |
| P (mm) | 329.8 | 357.1 | 384.1 | 330.2 | 199.8 |
| $\theta_r$ (m$^3$ m$^{-3}$) | _ | 0.077 | 0.077 | 0.076 | 0.064 |
| GWL (m) | _ | -3.8 | -3.6 | -3.0 | -3.5 |

Table 32. Typical values of total evapotranspiration ($E_T$, mm), total potential evapotranspiration ($E_{TP}$, mm), the indicator of land use/cover change ($D_{lu}$), the soil water stress of the root zone ($f_s$) and normalized $E_T$ ( $f_v$ (= $E_T/(E_{TP} \times f_s)$)) in the growing season of each period ( because there were some missing data in Period IV (from 12$^{th}$ September 2014 to 23$^{th}$ November 2014 and from 13$^{th}$ March 2015 to 22$^{th}$ April 2015), we removed the values of $E_T$, $E_{TP}$ and $f_s$  in these two time ranges  of Periods I-III).

| Items | $E_T$ | $E_{TP}$ | $D_{lu}D_{lu}$ | $f_s$ | $f_v$ |
|---|---|---|---|---|---|
| Periods | (mm) | (mm) | (%) | (dimensionless) | (dimensionless) |
| I | 238.4 | 876.1 | -0.2 | 0.62 | 78.1 |
| Growing  II | 236.5 | 870.7 | 4.6 | 0.63 | 79.9 |
| season  III | 292.1 | 956 | 10.4 | 0.59 | 86.3 |
| IV | 332.2 | 937 | 6 | 0.37 | 111.9 |

Fig. 1

[Figure]

Fig. 2

[Figure]

Fig. 3

[Figure]

Fig. 4

[Figure]

Fig. 5

[Figure]

Fig. 6

[Figure]

Fig. 7

[Figure]

**EDITORIAL CERTIFICATE**

This document certifies that the manuscript listed below was edited for proper English language, grammar, punctuation, spelling, and overall style by one or more of the highly qualified native English speaking editors at American Journal Experts.

**Manuscript title:**

Monitoring the variations of evapotranspiration due to the land use/cover changes in a semiarid shrubland

**Authors:**

Tingting Gong, Huimin Lei, Dawen Yang, Yang Jiao, Hanbo Yang

**Date Issued:**

January 5, 2017

**Certificate Verification Key:**

C015-9F4C-0BFC-F54E-5A90

[Figure]

This certificate may be verified at www.aje.com/certificate. This document certifies that the manuscript listed above was edited for proper English language, grammar, punctuation, spelling, and overall style by one or more of the highly qualified native English speaking editors at American Journal Experts. Neither the research content nor the authors' intentions were altered in any way during the editing process. Documents receiving this certification should be English-ready for publication; however, the author has the ability to accept or reject our suggestions and changes. To verify the final AJE edited version, please visit our verification page. If you have any questions or concerns about this edited document, please contact American Journal Experts at support@aje.com.

American Journal Experts provides a range of editing, translation and manuscript services for researchers and publishers around the world. Our top-quality PhD editors are all native English speakers from America's top universities. Our editors come from nearly every research field and possess the highest qualifications to edit research manuscripts written by non-native English speakers. For more information about our company, services and partner discounts, please visit www.aje.com.